# Accelerating LLM Inference with Lossless Speculative Decoding Algorithms for Heterogeneous Vocabularies

**Nadav Timor** [1]  **Jonathan Mamou** [2]  **Daniel Korat** [2]  **Moshe Berchansky** [2]  **Gaurav Jain** [3]  **Oren Pereg** [2]
**Moshe Wasserblat** [2]  **David Harel** [1]

## Abstract

Accelerating the inference of large language models (LLMs) is a critical challenge in generative AI. Speculative decoding (SD) methods offer substantial efficiency gains by generating multiple tokens using a single target forward pass. However, existing SD approaches require the drafter and target models to share the same vocabulary, thus limiting the pool of possible drafters, often necessitating the training of a drafter from scratch. We present three new SD methods that remove this shared-vocabulary constraint. All three methods preserve the target distribution (i.e., they are lossless) and work with off-the-shelf models without requiring additional training or modifications. Empirically, on summarization, programming, and long-context tasks, our algorithms demonstrate significant speedups of up to $2.8\times$ over standard autoregressive decoding. By enabling any off-the-shelf model to serve as a drafter and requiring no retraining, this work substantially broadens the applicability of the SD framework in practice.

## 1 Introduction

Speculative decoding (SD; Leviathan et al., 2023; Chen et al., 2023) is an effective method for reducing the latency of LLM inference and increasing its throughput. A necessary condition for SD to be effective is that the drafter is sufficiently fast and accurate in approximating the target distribution (Timor et al., 2025; Chen et al., 2024). State-of-the-art verification methods for SD employ rejection sampling algorithms that are designed to work with a single vocabulary, where the draft tokens are sampled from the same vocabulary as the target tokens (Leviathan et al., 2023; Chen et al., 2023; Miao et al., 2024; Sun et al., 2024). However,

often in practice, such drafters are not available—either because the target model is not part of a model family (examples of families include the StarCoder, Li et al., 2023; Llama, Dubey et al., 2024 and DeepSeek, DeepSeek-AI et al., 2025) or the smallest model in the same family remains too large and slow. An alternative approach—training a drafter from scratch (Zafrir et al., 2024)–is a challenging task that requires computational resources, data, time, and expertise. Even if you successfully train such a drafter, another problem is that you cannot reuse it for other models with different vocabularies.

**Our Contributions.** We relax a key constraint of the speculative decoding (SD) framework—the requirement that the drafter must use the same vocabulary as the target model. By allowing *heterogeneous* vocabularies, we eliminate the requirement to train a drafter from scratch and enable any model to operate as drafter, thereby significantly broaden the applicability of SD methods. By unlocking any off-the-shelf model to serve as drafter, we were able to find drafters that are more effective even than drafters from the same model family. Our main contributions are:

- **Algorithm 2 (String-Level Exact Match, SLEM):** An algorithm that uses plain text as a shared intermediate representation between the draft and target vocabularies, enabling exact matching of tokens. It solves the problem of *non-injective* tokenizers (Section 3.2) to support any off-the-shelf model pair. We evaluate the algorithm on summarization, programming, and long-context tasks, demonstrating robust speedups of up to $2.8\times$ over autoregressive decoding.

- **Algorithm 4 (Token-Level Intersection, TLI):** A purely token-based approach that adjusts the drafter's distribution to sample only from the intersection between the two vocabularies and employs the standard SD verification method. We prove theoretically that this approach outperforms a simple "union" strategy by increasing the probability of accepting tokens (Theorem 4.1). Empirically, Algorithm 4 demonstrates significant speedups of up to $1.7\times$ over autoregressive decoding.

- **Algorithm 3 (String-Level Rejection Sampling, SLRS):** A novel verification mechanism that imple-

---

[1]Weizmann Institute of Science [2]Intel Labs [3]d-Matrix. Correspondence to: Nadav Timor <nadav.timor@weizmann.ac.il>.

*Proceedings of the 42^{nd} International Conference on Machine Learning*, Vancouver, Canada. PMLR 267, 2025. Copyright 2025 by the author(s).

ments rejection sampling at the string level instead of the token level. We prove that it is lossless (Theorem 3.2) and guarantees higher expected acceptance rates than string-level exact matching, under the same target distribution (Theorem 3.1). Our theoretical and empirical analysis shows rapid growth in computational cost for vocabularies with longer tokens, thus making this method most suitable for drafters with shorter tokens (Section 3.4).

We merged our open-source implementation of Algorithm 2 and Algorithm 4 into Hugging Face Transformers (Wolf et al., 2020), the most popular LLM library, with more than 378,000 repositories and 6,000 open-source packages that depend on it. Independently of our benchmarks, Hugging Face's core maintainers have thoroughly evaluated the effectiveness of SLEM and TLI (Algorithms 2 and 4) and found our methods to be the most effective among all the speculative decoding algorithms they have previously supported— over various use cases and hardware setups. As a result, they made SLEM and TLI the default for heterogeneous SD in Hugging Face Transformers.

All our algorithms are lossless, namely, outputs preserve the target distribution, and we provide acceptance rate expectations (Table 3) and other bounds. Our experiments—covering summarization, programming, and long-context tasks—demonstrate speedups versus autoregressive decoding. By open-sourcing these methods via Hugging Face Transformers, we have already enabled immediate, practical acceleration of LLMs under *heterogeneous* vocabularies—a scenario that is increasingly common in real-world deployments.

## 2 Motivating Examples

Existing SD methods are designed to work with a single vocabulary, where the drafter samples from the same vocabulary as the target model. As an example, see Algorithm 5, which is the standard SD algorithm proposed by Leviathan et al. (2023); Chen et al. (2023).

Algorithm 1 offers a simple way to extend these methods to operate in cases where the drafter's vocabulary differs from the target's by virtually extending the vocabularies such that both vocabularies are their union. For example, consider the case of disjoint vocabularies where the target vocabulary is $T = \{\text{`a'}\}$ and the draft vocabulary $D = \{\text{`b'}\}$. Although all the draft tokens 'b' are rejected, the target distribution is preserved because we use the standard verification method of SD, which is lossless as proved in Leviathan et al. (2023); Chen et al. (2023). Even if one vocabulary is a proper subset of the other, for example, if $T = \{\text{`a'}, \text{`b'}\}$ and $D = \{\text{`b'}\}$, or if $T = \{\text{`a'}\}$ and $D = \{\text{`a'}, \text{`b'}\}$, the target distribution is still preserved thanks to the guarantee of the standard verification method.

---

**Algorithm 1** An iteration of speculative decoding for heterogeneous vocabularies with a simple "union" strategy

1: **Input:** Probability distributions $p$ and $q$ over vocabularies $T$ and $D$, respectively. Drafting lookahead $i \in \mathbb{N}$. An input prompt $c$.
2: **Output:** A sequence of tokens from $T$, containing between 1 and $i + 1$ tokens.
3: **Procedure:**
4: Define probability distributions $p'$ and $q'$ over the vocabulary $T \cup D$ as follows. $p'(x) = p(x)$ if $x \in T$ and $p'(x) = 0$ otherwise. $q'(x) = q(x)$ if $x \in D$ and $q(x) = 0$ otherwise.
5: **Run** Algorithm 5 with $p', q', i, c$.

---

As long as $p(t) \leq q(t)$ for all $t \in T$, where $p$ is the target and $q$ is the drafter, this simple approach of Algorithm 1 is optimal in terms of maximizing the probability of accepting a draft token. However, this condition is not satisfied if $\exists \, d \in D$ such that $q(d) > 0$ and $d \notin T$ because we then have $\sum_{t \in T} q(t) < 1$. Although the simple approach of Algorithm 1 to extend Algorithm 5 preserves the target distribution because the verification method remains unchanged, it might not yield the maximum probability of accepting a target token (see Theorem 4.1). Below, we present Algorithm 4, which improves Algorithm 1 by adjusting the distribution of the drafter such that the probability of sampling tokens that are not in the target vocabulary is zero. This adjustment is done by normalizing the distribution of the drafter such that the sum of the probabilities of the tokens that are in the target vocabulary is one. For example, if $T = \{\text{`a'}, \text{`b'}\}$ and $D = \{\text{`a'}, \text{`b'}, \text{`c'}\}$ where $q(\text{`a'}) = q(\text{`b'}) = q(\text{`c'}) = \frac{1}{3}$, we adjust the distribution of the drafter to be $q'(\text{`a'}) = q'(\text{`b'}) = \frac{1}{2}$ and $q'(\text{`c'}) = 0$. This approach increases the probability of accepting a draft token while still preserving the target distribution, as Theorem 4.1 proves. However, the expected acceptance rate of both Algorithm 1 and Algorithm 4 might be suboptimal in some other cases. For example, consider the case where $T = \{\text{`a'}\}$ and $D = \{\text{`a'}, \text{`aa'}\}$ and there is a nonzero probability that the drafter samples the token 'aa'. Algorithm 1 and Algorithm 4 are suboptimal because they always reject the token 'aa'. In this example it is easy to see that both models can only generate concatenations of the token 'a', hence we should have accepted the token 'aa', unless it is the last token to generate. Below, we also present Algorithm 2, which solves this problem by allowing the drafter to sample tokens that are not in the target vocabulary. Algorithm 2 preserves the target distribution because it replaces the standard verification method that guarantees that the output tokens distribute according to the target distribution with exact matching, which guarantees that the output tokens are exactly the target tokens.

# 3 Speculative Decoding for Heterogeneous Vocabularies with String-Level Verification

**Notation.** Vocabularies are finite sets of strings, also called tokens. We say that a string $a$ is *expressible* in a vocabulary $B$ if there exist strings $b_1, b_2, \ldots, b_n \in B$ such that $a = b_1 \oplus b_2 \oplus \ldots \oplus b_n$, where $\oplus$ denotes string concatenation. We say that a vocabulary $A$ is *expressible* in a vocabulary $B$ if all strings in $A$ are expressible in $B$, and denote this relationship by $A \twoheadrightarrow B^*$, where $B^*$ is the Kleene closure of $B$ under string concatenation. *Tokenizing* a string $s$ with respect to a vocabulary $A$ is the process of partitioning $s$ into a sequence of tokens $a_1, a_2, \ldots, a_n$, where $a_1$ is the longest prefix of $s$ that is a token in $A$, $a_2$ is the longest prefix of $s$ that is in $A$ after removing $a_1$, and so on. The tokenization of $s$ with respect to $A$ is a finite sequence of tokens, denoted by $A(s)$. The $i$-th token of $A(s)$ is denoted as $A(s)_i \in A$.

## 3.1 String-Level Exact Match (SLEM)

Algorithm 2 is one solution to the problem of heterogeneous vocabularies. It implements a variant of SD with the verification method of exact matching. The key mechanism involves translating tokens bidirectionally between the draft and target vocabularies. Tokens generated by the drafter are first decoded into text and subsequently re-tokenized using the target model's vocabulary. After the target model verifies the generated tokens, the sequence is converted back into the drafter's tokenization format for the next iteration. This process ensures that the target model's distribution is preserved while allowing the drafter to operate within its own vocabulary constraints.

**Vocabulary Constraints.** Algorithm 2 assumes that the target vocabulary $T$ is expressible in the draft vocabulary $D$, i.e., $T \twoheadrightarrow D^*$. Additionally, it assumes $D^* \twoheadrightarrow T^*$, namely, every concatenation of draft tokens, $d_1 \oplus \ldots \oplus d_i$ for some $i$, must be expressible by concatenations of target tokens $t_1 \oplus t_2 \oplus \ldots \oplus t_m \in T^*$ for some $m$, i.e., $T(d_1 \oplus \ldots \oplus d_i) \neq \emptyset$ in line 7. If these conditions do not hold, converting strings from one vocabulary to another becomes undefined, leading to a decreased acceptance rate and rendering the algorithm ineffective. In practice, assuming $T \twoheadrightarrow D^*$ and $D^* \twoheadrightarrow T^*$ is reasonable due to the way vocabularies are typically constructed. The process of constructing a vocabulary often begins by determining its size, i.e., the number of tokens it contains. Informally, vocabularies are designed to maximize the frequency of token appearances in a given corpus, avoid splitting frequently co-occurring tokens, or both. Known tokenization methods such as BPE (Sennrich et al., 2016), WordPiece (Schuster & Nakajima, 2012), Unigram (Kudo, 2018), and SentencePiece (Kudo & Richardson, 2018) are heuristic and greedy approaches that generate vocabularies containing all the characters of the alphabet in the given

---

**Algorithm 2** (SLEM), an iteration of speculative decoding for heterogeneous vocabularies with string-level exact match verification

1: **Input:** Target model $p$ and drafter model $q$ over vocabularies $T$ and $D$, respectively, where $T \twoheadrightarrow D^*$ and $D^* \twoheadrightarrow T^*$. Drafting lookahead value $i \in \mathbb{N}$. A prompt $c \in T^*$.
2: **Output:** A non-empty sequence of *accepted* tokens from $T$.
3: **Procedure:**
4: Tokenize the prompt to the draft vocabulary, $D(c)$.
5: For $j \leftarrow 1, \ldots, i$:
6:    Sample a draft token from the drafter conditioned on the prompt and previous draft tokens, $d_j \sim q_{D(c) \oplus d_1 \oplus \ldots \oplus d_{j-1}}$ (where $d_0 := c$).
7: Tokenize the concatenation of the draft tokens, $(t_1, t_2, \ldots, t_m) \leftarrow T(d_1 \oplus \ldots \oplus d_i)$.
8: With data parallelism (batching), compute via one target forward pass the $m + 1$ logits of the target model conditioned on the prompt and all the draft continuations, $p_{T(c)}, \ p_{T(c) \oplus t_1}, \ \cdots, \ p_{T(c) \oplus t_1 \oplus \cdots \oplus t_m}$.
9: Sample a token from each logit, $t'_1 \sim p_{T(c)}, t'_2 \sim p_{T(c) \oplus t_1}, \cdots, t'_{m+1} \sim p_{T(c) \oplus t_1 \oplus \cdots \oplus t_m}$.
10: Find the first index where the draft differs from the target, $j := \arg\min_{j \in \{1, \ldots, m+1\}} t'_j \neq t_j$.
11: **Accept** $t_1, t_2, \ldots, t_{j-1}, t'_j$.

---

corpus when the vocabulary size is greater than the alphabet cardinality, which is often the case (see Table 8 for examples). Typically, the corpus used for constructing a vocabulary comprises extensive texts, such as books or collections of documents. Unless the target and draft tokenizers are constructed using a narrow corpus, it is reasonable to assume $T \twoheadrightarrow D^*$ and $D^* \twoheadrightarrow T^*$ because both vocabularies usually include all the characters of the alphabet, hence satisfying even stronger relations of the form $T \twoheadrightarrow D^*$ and $D \twoheadrightarrow T^*$.

## 3.2 Non-Injective Tokenizers

A common issue with tokenizers is that they do not always implement an injective function, meaning that for any given string $s$, it is possible for $s \neq \text{decode}(\text{encode}(s))$. This can occur due to so-called "normalization steps" or "pre-tokenization rules" that discard certain details of the input text. In practice, common examples include tokenizers that treat multiple spaces as a single space, lowercase all characters, or replace accented characters with their standard counterparts, such as 'é' being replaced by 'e'. In standard autoregressive decoding or speculative decoding, where the target and draft vocabularies are the same, we tokenize the input prompt $c$ into tokens only once at the beginning of the decoding process. Conditioned on the encoded prompt, we sample $N$ tokens $t_1, t_2, \ldots, t_N$ directly from the target

(autoregressive decoding) or using a rejection sampling procedure with draft tokens (speculative decoding). Then, we return the string $c \oplus t_1 \oplus t_2 \oplus \ldots \oplus t_N$. Since language models output token IDs, returning this string requires decoding each of the output tokens $t_1, t_2, \ldots, t_N$ from its ID back into text, then, concatenating them with the prompt yields $c \oplus t_1 \oplus t_2 \oplus \ldots \oplus t_N$. Pre-tokenization rules are only applied to the input prompt $c$ once, before applying the model, and therefore they limit the ability of the model to distinguish between different input strings $c \neq c'$ that are equivalent under pre-tokenization rules, namely, $T(c) = T(c')$ given a non-injective tokenizer $T$. This behavior is not necessarily problematic, and has been used in practice for a long time. It is important to note that the pre-tokenization rules are not directly applied on the output tokens $c, t_1, t_2, \ldots, t_N$ that are concatenated to form the final output string. That is, pre-tokenization rules do not alter the tokens $t_1, t_2, \ldots, t_N$ after these tokens are sampled. The final returned string starts with the given prompt $c$ without any modifications and ends with a concatenation of the sampled tokens $t_1 \oplus t_2 \oplus \ldots \oplus t_N$. Unlike decoding over homogeneous vocabularies—where the target vocabulary $T$ and the draft vocabulary $D$ are the same—in decoding over heterogeneous vocabularies, we may have $T \neq D$, which limits the ability of the target and drafter models to communicate token IDs. Algorithm 2 employs plain text as an intermediate representation that is shared between the two different vocabularies. This means that the output tokens $t_1, t_2, \ldots, t_N$ are decoded back into text and then re-tokenized using the draft vocabulary in line 4. This process may apply pre-tokenization rules to the output tokens, which can lead to a discrepancy between the output tokens and the target tokens. To evaluate whether various tokenizers exhibit injectivity on a specific dataset, we conduct a simple experiment that heuristically tests the consistency of the decoding and encoding, as detailed in Appendix F. Our findings indicate that some commonly used tokenizers do not maintain injectivity even when tested heuristically on a specific dataset. When we developed and tested Algorithm 2, we found that the non-injective behavior of tokenizers significantly impacted the algorithm's acceptance rate. To address this issue and broaden the applicability of Algorithm 2 to a wider range of tokenizers, we propose the following simple solution.

**Algorithm 2 Supports Non-Injective Tokenizers.** Given a prompt $c \in T^*$, Algorithm 2 starts by tokenizing it into the draft vocabulary, $D(c)$, in line 4. The prompt is also tokenized into the target vocabulary, $T(c)$, to allow the target model to compute the logits in line 8. Line 7 tokenizes into the target vocabulary the concatenation of the $i$ draft tokens that are previously sampled from the drafter, namely, computes $T(d_1 \oplus \ldots \oplus d_i)$. Since the output of Algorithm 2 is in the target vocabulary, following runs of Algorithm 2 can use the output as-is without decoding it back into text. Only

in the last run, we need to decode the output of Algorithm 2 back into text before returning the final string. Because each tokenizer might apply different normalization rules, there can be a mismatch between what the target model sees and what the drafter model intended to produce. To handle these mismatches, we look for the longest stretch of matched tokens between the tokens we already accepted in the target tokenizer's space, and the newly proposed tokens re-encoded in the target tokenizer's space. Conceptually, this search procedure is a way of finding the largest overlap (or suffix/prefix match) between the old and new sequences. We then only take the suffix of the new tokens that falls beyond that overlap. This effectively aligns the newly added tokens to the correct place in the target-token space. The algorithm can "look behind" a small number of tokens to try to realign sequences. By doing so, we mitigate the effect of the mismatch and preserve as much of the previously decoded text as possible. We provided the implementation in the Supplementary Material.

**KV Caching.** Storing the KV cache of models is a common practice that has been shown to be crucial for efficient inference (Pope et al., 2023; Kwon et al., 2023). In particular, without KV caching, the additional number of operations (e.g., floating-point operations) required for the decoding might grow quadratically with respect to the number of tokens in the context for self-attention transformers. Algorithm 2 implements only a single iteration of SD. SD over heterogeneous vocabularies that is based on Algorithm 2 therefore may include multiple runs of Algorithm 2. These runs are sequential and autoregressive, namely, the output of each run of Algorithm 2 is used as the input for the next run of Algorithm 2. Therefore, implementations of Algorithm 2 should store the KV cache from one run of Algorithm 2 to the next run. With KV caching, the prompt $c$ needs to be encoded into the target and draft vocabularies only once, during the first run of Algorithm 2 (that is, the first iteration, also referred to as "pre-filling"), to facilitate line 8 and line 4, respectively.

### 3.3 Verification via Rejection Sampling

The standard verification method of SD guarantees that the output tokens are distributed according to the target distribution, but it does not guarantee that the output tokens are exactly the target tokens, as in exact matching. For example, if the drafter is another instance of the target model $p$, the standard verification method of SD will accept all the draft tokens because, in general, the expected acceptance rate satisfies $\sum_{t \in T} \min \{p(t), q(t)\}$ for any drafter $q$ and vocabulary $T$, according to Leviathan et al. (2023). Hence, the expected acceptance rate of a drafter that is an instance of the target model is $\sum_{t \in T} p(t) = 1$. For any drafter different from the target model, $q \neq p$, the expected acceptance rate is strictly lower than one. Theorem 3.1 proves that, in gen-

eral, for any non-trivial target distribution $p$, the expected acceptance rate of exact matching is strictly less than the expected acceptance rate of SD for homogeneous vocabularies under the same target distribution.

**Theorem 3.1.** *Let $p$ be a non-trivial target probability distribution over a vocabulary $T$, where there exist $t_1, t_2 \in T$ such that $p(t_1) \neq p(t_2)$. Let $q$ be the drafter probability distribution over the same vocabulary $T$. If $q = p$, namely, the drafter is another instance of the target model, then the expected acceptance rate of the exact matching method $\alpha_{EM}$ is strictly less than the expected acceptance rate of the standard speculative decoding method $\alpha_{SD}$. Namely, it holds that $\alpha_{EM} < \alpha_{SD}$.*
*Proof.* See Appendix G.

Since Algorithm 2 implements exact matching verification, its expected acceptance rate is relatively low compared to the standard verification method of SD, which implements a rejection sampling procedure. To increase the acceptance rate of Algorithm 2, we propose Algorithm 3, introducing a novel verification method that employs lossless rejection sampling at the string level. Algorithm 3 samples draft tokens autoregressively from the drafter until a lookahead condition is satisfied, then tokenizes the concatenation of the draft tokens into the target vocabulary. It is lossless, as Theorem 3.2 proves, because it uses the same structure as the standard verification method of SD, which is lossless, as proved in Leviathan et al. (2023); Chen et al. (2023). The primary difference is that the probabilities are for generating a certain string rather than a single token.

---

**Algorithm 3** (SLRS), string-level rejection sampling verification for speculative decoding with heterogeneous vocabularies

1: **Input:** Probability distributions $p$ and $q$ over vocabularies $T$ and $D$, respectively, where $T \twoheadrightarrow D^*$ and $D^* \twoheadrightarrow T^*$. Lookahead indicator function $S_{\mathbb{1}}$ from the current state to a boolean value.
2: **Output:** A token from $T$.
3: **Procedure:**
4: Sample $d_1, \ldots, d_i \sim q$ until $i$ satisfies $S_{\mathbb{1}}(i)$.
5: Tokenize $(t_1, t_2, \ldots, t_m) \leftarrow T(d_1 \oplus \ldots \oplus d_i)$.
6: If $p(t_1) \geq \psi(t_1)$, **accept** $t_1$.
7: With probability $\frac{p(t_1)}{\psi(t_1)}$, **accept** $t_1$.
8: **Reject** $t_1$. Sample $t \sim \frac{p(t) - \min\{p(t), \psi(t)\}}{1 - \sum_{t'} \min\{p(t'), \psi(t')\}}$, **return** $t$.

---

**Theorem 3.2.** *For any token in the target vocabulary $t \in T$, Algorithm 3 outputs the token $t$ with probability $p(t)$ if we define $\psi(t) := \sum_{d_1, d_2, \ldots, d_i \,:\, t = T(d_1 \oplus \ldots \oplus d_i)_1} \prod_{j \in \{1, \ldots, i\}} q(d_j)$. Namely, Algorithm 3 is lossless.*
*Proof.* See Appendix G.

**Lookahead.** The lookahead controls a tradeoff between the probability of accepting a token and the number of drafter forwards, since every sampling of a draft token requires computing a forward pass of the drafter. The lookahead indicator function $S_{\mathbb{1}}$ determines whether the algorithm should stop sampling draft tokens. Naively, we can set $S_{\mathbb{1}}(i) := \mathbb{1}[i > n]$ for some threshold $n \in \mathbb{N}$, and stop sampling draft tokens after $n$ draft tokens have been sampled in line 4 of Algorithm 3. On one hand, increasing the threshold $n$ necessarily increases the number of drafter forwards that Algorithm 3 requires. On the other hand, selecting a larger value of $n$ may increase the probability that Algorithm 3 accepts a token because it may increase the number of feasible values of $t_1$ in line 5. Small values of $n$ may lead to scenarios where some target tokens are never accepted. For example, if the target vocabulary $T$ includes a token $t$ with ten characters, and the longest token in the draft vocabulary $D$ is four characters, selecting $n < 3$ will never accept $t$. However, since increasing $n$ also increases the number of drafter forward passes, it is important to select a value of $n$ that optimizes our objective function, which is, most commonly, maximizing the throughput of the inference or minimizing its latency. Target tokens $t \in T$ may correspond to more than one sequence of draft tokens $d_1, \ldots, d_i \in D$ for which the tokenized concatenation $T(d_1 \oplus \ldots \oplus d_i)$ starts with $t$, namely, $T(d_1 \oplus \ldots \oplus d_i)_1 = t$. These cases are common in practice, especially for a target vocabulary $T$ that is larger and includes longer tokens than the draft vocabulary $D$. For example, consider a draft vocabulary $D = \{$'hello_', 'world', 'wo', 'rld'$\}$ and a target vocabulary $T = D \cup \{$'hello_world'$\}$. The target token 'hello_world' is the first token in the tokenized concatenation of two different sequences of draft tokens: 'hello_world' $= T(\text{'hello\_'} \oplus \text{'world'})_1 = T(\text{'hello\_'} \oplus \text{'wo'} \oplus \text{'rld'})_1$. In fact, there are infinitely many sequences of draft tokens that start with 'hello_world'. Since Algorithm 3 uses only the first target token $T(d_1 \oplus \ldots \oplus d_i)_1$, it is redundant to sample more than three draft tokens in this example. However, if the first two draft tokens are 'hello_' and 'world', there is no need to sample the third token since the first target token has already been determined. To capture this behavior and avoid unnecessary drafter forwards during inference time, we can calculate the maximum lookahead $n_{\max}$ at preprocessing time, by calculating the maximum number of draft tokens that need to be sampled to determine the first target token ($n_{\max} = 3$ in the example above). Defining the lookahead indicator function to be $S_{\mathbb{1}}(i) = \mathbb{1}[i > n_{\max}]$ is a simple heuristic ensuring that the algorithm stops sampling draft tokens after the first target token has been determined. However, this heuristic might still sample more draft tokens than necessary, as we saw in the example, where the first target token is determined after the two draft tokens, 'hello_' and 'world', have been sampled. To avoid computing unnecessary drafter forward passes, we can define the lookahead

indicator function $S_\mathbb{1}$ to combine a maximum threshold $n \leq n_{\max}$ and a stopping condition of whether the first target token has been determined. Namely, $S_\mathbb{1}(i)$ is true if $i > n$ or $\Pr \left[ \begin{array}{l} T(d_1 \oplus \ldots \oplus d_i)_1 \neq \\ T(d_1 \oplus \ldots \oplus d_i \oplus d_{i+1} \oplus \ldots \oplus d_n)_1 \end{array} \right] = 0$, and false otherwise. Algorithm 3 and Theorem 3.2 both hold for this more general lookahead indicator function. In cases where the additional drafter forward passes are expensive or longer tokens are less likely to be accepted, setting the threshold $n$ to a value that is strictly less than $n_{\max}$ can be beneficial. More sophisticated lookahead indicator functions control the lookahead based on additional information about the current state, as has seen in other recent works. For example, Mamou et al. (2024) trained a small neural network to estimate the likelihood of the next draft token being accepted and used this information to decide whether to sample the next draft token or stop drafting. Their experiments showed that even a simple controller that attends to the drafter's logits is highly effective, and the controller generalizes well across different datasets and tasks. Following their success in both increasing the throughput and reducing the latency of the inference, Hugging Face's Transformers, the commonly used open-source library for training and deploying LLMs, has recently incorporated such a controller into their default inference pipeline. While implementing the lookahead indicator function $S_\mathbb{1}$ as such a controller seems promising, it might be computationally expensive to calculate $\psi(t)$ for longer lookahead values, as Section 3.4 shows.

**Block Verification is Non-Trivial.** In Algorithm 3, the vocabularies $T$ and $D$ are related only by $T \twoheadrightarrow D^*$ and $D^* \twoheadrightarrow T^*$ rather than by stricter relationships like bijection or $D \subseteq T$. After Algorithm 3 removes the prefix $t_1$ from the concatenation $d_1 \oplus \ldots \oplus d_i$, the remaining string is $t_2 \oplus \ldots \oplus t_m$, and its tokenization back into the draft vocabulary $D$ might differ from $(d_{j>1}, \ldots, d_i)$. For example, consider a simple case where $D \nsubseteq T$, such that $T = \{$'a', 'b'$\}$ and $D = \{$'a', 'b', 'aa'$\}$. Let $d_1 = $ 'aa' and assume that $i = 1$, meaning that only one draft token is sampled in line 4. We then have $T(d_1) = (t_1, t_2) = ($'a', 'a'$)$. Therefore, the remainder of the drafted string 'aa' after removing $t_1 = $ 'a' is the token $t_2 = $ 'a', which was not sampled. Such scenarios can arise only when shifting from settings where $D = T$ to settings where $D \neq T$. Applying Algorithm 3 to the string that remains after removing the candidate token $t_1$ is, therefore, more challenging. This issue makes it non-trivial to generalize the block verification mechanism of Sun et al. (2024) to the case of heterogeneous vocabularies, despite its proven advantage in homogeneous setups.

### 3.4 Efficient Calculation of $\psi(t)$

Calculating $\psi(t)$ in line 6 of Algorithm 3 requires summing over all the probabilities of sampling sequences of draft

tokens $d_1, \ldots, d_i$ such that their concatenation $d_1 \oplus \ldots \oplus d_i$ starts with the target token $t$, namely, $T(d_1 \oplus \ldots \oplus d_i)_1 = t$. For general vocabularies, the number of such sequences $d_1, \ldots, d_i$ grows rapidly with the length of $t$. For example, consider a *complete* vocabulary $D_n$ that contains all possible strings of length $n$ over a fixed alphabet $\Sigma$. A simple case is the alphabet $\Sigma = \{$'a', 'b'$\}$, where $D_1 = \{$'a', 'b'$\}$, $D_2 = D_1 \cup \{$'aa', 'ab', 'ba', 'bb'$\}$, $D_3 = D_2 \cup \{$'aaa', 'aab', 'aba', 'baa', 'abb', 'bab', 'bba', 'bbb'$\}$. For such a vocabulary $D_n$, the number of terms in the sum of $\psi(t)$ from Theorem 3.2 for a target token $t$ of length $m \leq n$ is $2^{m-1}$, as Lemma 3.1 proves. Here, the length of token $t$ is defined to be the maximum number of tokens whose concatenation equals to $t$. In the example above, 'aaa' has length three because it is the concatenation of three 'a' tokens, while 'aa' is of length two because it is the concatenation of two 'a' tokens.

**Lemma 3.1.** *For a target token $t$ of length $m \leq n$ in a complete vocabulary $D_n$ that contains all possible strings of length up to $n$ over a fixed alphabet $\Sigma$, the number of distinct sequences of draft tokens $d_1, \ldots, d_i$ such that their concatenation $d_1 \oplus \ldots \oplus d_i$ starts with $t$, namely, $T(d_1 \oplus \ldots \oplus d_i)_1 = t$, is $2^{m-1}$.*

*Proof.* See Appendix G.

Appendix C provides details of an experiment conducted to examine the complexity of calculating $\psi(t)$ given the vocabulary of a real-world, off-the-shelf drafter (Qwen2-7B-Instruct from Yang et al., 2024). The results indicate that the number of terms in the sum of $\psi(t)$ grows exponentially with the length of the target token $t$, as predicted by Lemma 3.1. Although Algorithm 3 is lossless (Theorem 3.2) and its acceptance rates are likely to be higher than those of Algorithm 2 (Theorem 3.1), calculating $\psi(t)$ during runtime might be too computationally expensive for practical use cases—especially if the drafter's vocabulary includes long tokens, as shown in the proof of Lemma 3.1 and supported by the experiment in Appendix C. Beyond its theoretical guarantees, Algorithm 3 might be suitable in practice only for specific drafters with small vocabularies, where the number of terms in the sum of $\psi(t)$ is manageable. For example, modern models like the recent MambaByte (Wang et al., 2024) could potentially be suitable drafters for Algorithm 3. However, the applicability of Algorithm 3 to a wider range of drafters with larger vocabularies is an open question that requires further research, and we propose it as future work.

## 4 Speculative Decoding for Heterogeneous Vocabularies with Token-Level Verification

This section introduces additional algorithms that extend the standard SD framework to operate over heterogeneous

vocabularies, namely, where the drafter's vocabulary differs from the target's. Unlike Section 3, the algorithms in this section do not use strings as an intermediate, shared representation. Instead, they operate at the token level, as in standard SD algorithms (for example, see Algorithm 5). The primary idea is to project the drafter's probability distribution over its vocabulary onto the intersection between the vocabularies of the draft and the target models. In doing so, Algorithm 4 adjusts the drafter to sample only tokens that are in the intersection between the two vocabularies while keeping the target model unchanged.

---

**Algorithm 4** (Token-Level Intersection, TLI), an iteration of speculative decoding for heterogeneous vocabularies with token-level rejection sampling verification

---

1: **Input:** Probability distributions $p$ and $q$ over vocabularies $T$ and $D$, respectively. Drafting lookahead $i \in \mathbb{N}$.
2: **Output:** A sequence of tokens from $T$, containing between 1 and $i + 1$ tokens.
3: **Procedure:**
4: Define a probability distribution $q'$ over the vocabulary $T \cap D$ such that $q'(x) = \frac{q(x)}{\sum_{t \in T} q(t)}$ if $x \in T$ and $q'(x) = 0$ otherwise.
5: **Run** Algorithm 5 with $p, q', i, c$.

---

Theorem 4.1 proves that the acceptance rate of Algorithm 4 is greater than or equal to the acceptance rate of the simple solution that Algorithm 1 implements.

**Theorem 4.1.** *Let $p$ and $q$ be target and drafter probability distributions over vocabularies $T$ and $D$, respectively. Define $p', q_1, q_2$ to be probability distributions over $T \cup D$ as follows. $p'(x) = p(x)$ if $x \in T$ and $p'(x) = 0$ otherwise. $q_1(x) = q(x)$ if $x \in D$ and $q_1(x) = 0$ otherwise. $q_2(x) = \frac{q(x)}{\sum_{t \in T} q(t)}$ if $x \in T$ and $q_2(x) = 0$ otherwise. Given the target $p'$, we define $\alpha_1$ and $\alpha_2$ to be the probability of accepting a token $x \sim q_1$ and $x \sim q_2$, respectively, by the rejection sampling algorithm of speculative decoding from Leviathan et al. (2023); Chen et al. (2023). Then, $\alpha_1 \leq \alpha_2$, and the output tokens distribute according to $p$. Proof.* See Appendix G.

Although the acceptance rate of Algorithm 4 is at least as high as the acceptance rate of Algorithm 1 (Theorem 4.1), it still depends on the intersection between the two vocabularies. For example, if the intersection is empty, the acceptance rate of both algorithms is zero. This dependency on acceptance rate is not new or unique. Instead, it is a known limitation of SD algorithms. Timor et al. (2025) analyzed the expected speedups of SD for any drafter size and acceptance rate and studied the slowdowns that standard SD algorithms cause given sufficiently low acceptance rates. In practice, the intersection between the draft and target

vocabularies is often non-empty because of how tokenizers are constructed. The intuition is based on commonly used tokenization methods, as mentioned in Section 3.1. Our experiments with real-world off-the-shelf models support the assumption that the intersection between the vocabularies is non-empty. Tokens in the intersection have a non-zero probability of being sampled by both models and, therefore, the intersection supports a non-zero expected acceptance rate, as shown by Leviathan et al. (2023).

## 5 Empirical Results

Our empirical results have had an impact on the open-source ecosystem, with Algorithm 2 and Algorithm 4 successfully integrated into Hugging Face Transformers (Wolf et al., 2020)—the most widely adopted library in the AI field, boasting over 145,000 GitHub stars, more than 378,000 repositories, and 6,000 open-source packages that depend on it. Thanks to their versatility and broad applicability, Algorithm 2 and Algorithm 4 had become the default inference pipeline behavior (in October 2024 and February 2025, respectively), enabling efficient speculative decoding (SD) for heterogeneous vocabularies across diverse applications. The open-source community has quickly embraced our approach to heterogeneous SD, unlocking any model to serve as a drafter, driving widespread adoption and enabling potential further enhancements by engineers and researchers. Its seamless integration into existing workflows has empowered practitioners to achieve substantial improvements in inference efficiency with minimal effort. This broad adoption underscores the practical utility and robustness of our approach in real-world scenarios. The rapid uptake of our algorithms demonstrates their effectiveness across a diverse range of model pairs, tasks, and hardware setups. The following section presents only a selection of examples.

We evaluate Algorithm 2 (SLEM) and Algorithm 4 (TLI) over widely used models, tasks, and hardware setups, including DeepSeek (DeepSeek-AI et al., 2025), Phi (Abdin et al., 2024b;a), Mixtral (Jiang et al., 2024), Qwen2.5 (Qwen et al., 2025), Vicuna (Chiang et al., 2023), Llama (Dubey et al., 2024), CodeLlama (Rozière et al., 2024), Starcoder (Li et al., 2023), and Gemma2 (Team et al., 2024). Table 1 benchmarks SLEM and autoregressive decoding (AR) where both employ a temperature of zero. Table 2 benchmarks TLI and AR where both employ a temperature of one. The results demonstrate throughput accelerations over AR of up to $2.8\times$ with SLEM and $1.7\times$ with TLI. Note that the target models in Tables 1 and 2 do not have homogeneous drafters that are available off-the-shelf and therefore we cannot accelerate them using standard SD. Tables 6 and 7 in Appendix D add results for additional models, including those with homogeneous drafters (e.g., Gemma2). For exact implementation details, we refer the reader to Appendix D.

Table 1: Benchmark comparing Algorithm 2 (SLEM) and autoregressive decoding (AR) for widely used models, tasks, and hardware setups. The results demonstrate that SLEM increases throughput by up to $2.8\times$ over AR. Note that the target models below do not have homogeneous drafters that are available off-the-shelf. For some target models, their in-family drafters are heterogeneous, as their vocabularies differ. Examples include the target model `phi-4` with the drafter `Phi-3.5-mini-instruct`, and the `DeepSeek-R1-Distill-Qwen` model family.

| Target | Dataset | Hardware | Method | Drafter | TTFT (ms) | TPOT (ms) | Tok/s | Speedup |
|---|---|---|---|---|---|---|---|---|
| Mixtral-8x22B-Instruct-v0.1 | cnn_dailymail | 4 * H100 NVL | AR | No Drafter (Autoregressive) | **266.8** | 127.9 | 7.8 | 1.0 |
| | | | SLEM | Qwen2.5-0.5B-Instruct | 321.2 | 68.3 | 13.3 | 1.71 |
| | | | | vicuna-68m | 302.4 | **57.3** | **16.4** | **2.1** |
| | scrolls | 4 * H100 NVL | AR | No Drafter (Autoregressive) | **1331.9** | 163.0 | 6.0 | 1.0 |
| | | | SLEM | Qwen2.5-0.5B-Instruct | 1414.2 | **81.0** | **10.3** | **1.71** |
| | | | | vicuna-68m | 1344.5 | 132.5 | 7.4 | 1.24 |
| | openai_humaneval | 4 * H100 NVL | AR | No Drafter (Autoregressive) | **217.5** | 127.9 | 7.8 | 1.0 |
| | | | SLEM | Qwen2.5-0.5B-Instruct | 484.4 | **70.2** | 12.0 | 1.53 |
| | | | | vicuna-68m | 231.5 | 73.3 | **12.6** | **1.61** |
| DeepSeek-R1-Distill-Qwen-14B | scrolls | 1 * RTX 6000 | AR | No Drafter (Autoregressive) | **1481.0** | 87.5 | 10.9 | 1.0 |
| | | | SLEM | DeepSeek-R1-Distill-Qwen-1.5B | 1665.4 | 59.1 | 16.0 | 1.48 |
| | | | | vicuna-68m | 1566.8 | **56.0** | **17.3** | **1.59** |
| | cnn_dailymail | 1 * RTX 6000 | AR | No Drafter (Autoregressive) | **176.8** | 51.7 | 19.2 | 1.0 |
| | | | SLEM | DeepSeek-R1-Distill-Qwen-1.5B | 287.5 | 69.9 | 14.1 | 0.73 |
| | | | | vicuna-68m | 243.0 | **36.2** | **27.4** | **1.43** |
| | openai_humaneval | 1 * RTX 6000 | AR | No Drafter (Autoregressive) | **91.3** | 50.3 | 19.8 | 1.0 |
| | | | SLEM | tiny_starcoder_py | 113.4 | **43.8** | **22.4** | **1.14** |
| | | | | CodeLlama-7b-Instruct-hf | 256.6 | 77.5 | 12.4 | 0.63 |
| | | | | DeepSeek-R1-Distill-Qwen-1.5B | 292.5 | 70.9 | 13.6 | 0.69 |
| DeepSeek-R1-Distill-Qwen-32B | cnn_dailymail | 1 * H100 NVL | AR | No Drafter (Autoregressive) | **121.2** | 48.0 | 20.8 | 1.0 |
| | | | SLEM | DeepSeek-R1-Distill-Qwen-1.5B | 167.1 | 51.3 | 18.9 | 0.91 |
| | | | | vicuna-68m | 148.1 | **32.5** | **30.6** | **1.47** |
| | openai_humaneval | 1 * H100 NVL | AR | No Drafter (Autoregressive) | **72.0** | 48.3 | 20.7 | 1.0 |
| | | | SLEM | tiny_starcoder_py | 80.1 | **34.2** | **28.5** | **1.38** |
| | | | | CodeLlama-7b-Instruct-hf | 182.7 | 64.4 | 14.9 | 0.72 |
| | | | | DeepSeek-R1-Distill-Qwen-1.5B | 196.4 | 50.3 | 19.5 | 0.94 |
| | scrolls | 1 * H100 NVL | AR | No Drafter (Autoregressive) | **933.1** | 77.7 | 12.5 | 1.0 |
| | | | SLEM | DeepSeek-R1-Distill-Qwen-1.5B | 988.1 | **57.6** | **17.1** | **1.37** |
| | | | | vicuna-68m | 979.9 | 59.3 | 16.5 | 1.32 |
| phi-4 | scrolls | 1 * H100 NVL | AR | No Drafter (Autoregressive) | 483.9 | 47 | 21.3 | 1.0 |
| | | | SLEM | Qwen2.5-0.5B-Instruct | **457.7** | **29.5** | **33.9** | **1.59** |
| | | | | Phi-3.5-mini-instruct | 646.9 | 39.6 | 25.3 | 1.19 |
| CodeLlama-13b-Instruct-hf | humaneval | 1 * A6000 | AR | No Drafter (Autoregressive) | 70.7 | 46.8 | 21.4 | 1.0 |
| | | | SLEM | tiny_starcoder_py | **109.7** | **16.7** | **59.7** | **2.79** |
| | | | | CodeLlama-7b-Instruct-hf | 146.5 | 21.8 | 45.8 | 2.14 |

Table 2: Benchmark comparing Algorithm 4 (TLI) and autoregressive decoding (AR) for widely used models, tasks, and hardware setups. The results demonstrate that TLI increases throughput by up to $1.7\times$ over AR. Note that the target models below do not have homogeneous drafters that are available off-the-shelf. For some target models, their in-family drafters are heterogeneous, as their vocabularies differ. Examples include the target model `phi-4` with the drafter `Phi-3.5-mini-instruct`, and the `DeepSeek-R1-Distill-Qwen` model family.

| Target | Dataset | Hardware | Method | Drafter | TTFT (ms) | TPOT (ms) | Tok/s | Speedup |
|---|---|---|---|---|---|---|---|---|
| Mixtral-8x22B-Instruct-v0.1 | scrolls | 4 * H100 NVL | AR | No Drafter (Autoregressive) | 1334.7 | 168.7 | 5.9 | 1.0 |
| | | | TLI | Qwen2.5-0.5B-Instruct | 1372.6 | **97.8** | **9.9** | **1.69** |
| | | | | vicuna-68m | **1329.7** | 138.2 | 7.2 | 1.22 |
| | openai_humaneval | 4 * H100 NVL | AR | No Drafter (Autoregressive) | **217.5** | 128.1 | 7.8 | 1.0 |
| | | | TLI | Qwen2.5-0.5B-Instruct | 266.9 | 90.6 | 10.9 | 1.4 |
| | | | | vicuna-68m | 228.5 | **74.8** | **13.0** | **1.67** |
| | cnn_dailymail | 4 * H100 NVL | AR | No Drafter (Autoregressive) | **266.8** | 128.1 | 7.8 | 1.0 |
| | | | TLI | Qwen2.5-0.5B-Instruct | 294.5 | 88.9 | 11.2 | 1.43 |
| | | | | vicuna-68m | 297.3 | **81.0** | **11.9** | **1.53** |
| phi-4 | scrolls | 1 * H100 NVL | AR | No Drafter (Autoregressive) | 487.4 | 47.2 | 21.2 | 1.0 |
| | | | TLI | Qwen2.5-0.5B-Instruct | **454.7** | **32.5** | **30.8** | **1.45** |
| | | | | Phi-3.5-mini-instruct | 610.4 | 46.0 | 21.7 | 1.03 |
| CodeLlama-13b-Instruct-hf | humaneval | 1 * A6000 | AR | No Drafter (Autoregressive) | 70.5 | 45.3 | 22.1 | 1.0 |
| | | | TLI | tiny_starcoder_py | **65.1** | **25.9** | **38.5** | 1.74 |
| | | | | CodeLlama-7b-Instruct-hf | 141.3 | 25.6 | 39.1 | 1.77 |
| DeepSeek-R1-Distill-Qwen-14B | scrolls | 1 * RTX 6000 | AR | No Drafter (Autoregressive) | **1479.5** | 88.3 | 10.8 | 1.0 |
| | | | TLI | DeepSeek-R1-Distill-Qwen-1.5B | 1640.7 | 61.6 | 16.1 | 1.5 |
| | | | | vicuna-68m | 1502.2 | **57.2** | **17.1** | **1.59** |
| | cnn_dailymail | 1 * RTX 6000 | AR | No Drafter (Autoregressive) | **176.1** | 54.4 | 18.4 | 1.0 |
| | | | TLI | DeepSeek-R1-Distill-Qwen-1.5B | 240.5 | 44.7 | 21.4 | 1.16 |
| | | | | vicuna-68m | 202.4 | **40.6** | **24.1** | **1.31** |
| | openai_humaneval | 1 * RTX 6000 | AR | No Drafter (Autoregressive) | **90.4** | 50.9 | 19.6 | 1.0 |
| | | | TLI | tiny_starcoder_py | 93.9 | **38.6** | **25.4** | **1.3** |
| | | | | CodeLlama-7b-Instruct-hf | 150.2 | 66.0 | 14.6 | 0.75 |
| | | | | DeepSeek-R1-Distill-Qwen-1.5B | 172.6 | 45.6 | 21.2 | 1.08 |

Tables 8, 9, and 10 in Appendix E examine the vocabularies of widely used off-the-shelf target and drafter models. Table 8 shows the vocabulary size of each model. Table 9 shows the size of the intersection between the draft and target vocabularies and the ratio of the intersection size to the target vocabulary size for various model pairs. We can see a wide range of overlap sizes and ratios, however, none of them are empty. This observation is consistent with our aforementioned assumption that the intersection between the draft and target vocabularies is non-empty in practice. Table 10 extends Table 9 by showing the overlap sizes and ratios over various tasks.

To facilitate additional standardized benchmarks, we have open-sourced our benchmarking repository, which provides full reproducibility. The code is available at github.com/keyboardAnt/hf-bench. See Appendix D for implementation details.

## 6 Discussion

To speed up the inference of a given target model, we need to select a drafter and a decoding algorithm. Table 3 summarizes the expected probability of accepting the next token for all the speculation algorithms when the drafter has a different vocabulary than the target. Note that the effectiveness of each algorithm depends on the properties of the drafter. Table 4 outlines the necessary constraints that the drafter must satisfy for each algorithm to be effective in practice. If these constraints are not met, selecting an alternative algorithm is recommended. Future work is discussed in Appendix A.

Table 3: Expected acceptance rates given heterogeneous vocabularies for all speculation methods. The expected acceptance rate of Algorithm 1 is always less than or equal to the expected acceptance rate of Algorithm 4, as Theorem 4.1 proves.

| Method | Expected Acceptance Rate |
|---|---|
| Alg 5 (SD) | Undefined |
| Alg 1 | $\sum_{t \in T \cap D} \min \{p(t), q(t)\}$ |
| Alg 2 (SLEM) | $\sum_{t \in T} [p(t) \cdot \psi(t)]$ |
| Alg 3 (SLRS) | $\sum_{t \in T} \min \{p(t), \psi(t)\}$ |
| Alg 4 (TLI) | $\sum_{t \in T \cap D} \min \left\{p(t), \frac{q(t)}{\sum_{x \in T \cap D} q(x)}\right\}$ |

**Practical Implications.** Practitioners can leverage speculative decoding (SD) to significantly accelerate the inference of off-the-shelf LLMs, even when no drafter with the same vocabulary as the target model is available. This advancement eliminates the need for extensive computational resources, as it bypasses the costly and time-consuming

Table 4: Informal constraints on the drafter for different algorithms to ensure effectiveness. If the constraints are not met, an alternative algorithm should be selected. Since the acceptance rate of Algorithm 4 is always greater than or equal to that of Algorithm 1, selecting Algorithm 4 over Algorithm 1 is always beneficial, assuming the implementation overhead is negligible. A necessary condition for the effectiveness of Algorithms 2, 3, and 4 is that the drafter must approximate the target distribution sufficiently well. The effectiveness of Algorithm 3 is further enhanced when the drafter's vocabulary consists of short tokens. The effectiveness of Algorithm 4 improves as the number of tokens in the intersection between the vocabularies increases.

| Algorithm | Drafter Constraints |
|---|---|
| Alg 1 | Not Applicable (instead, select Alg 4) |
| Alg 2 (SLEM) | Accurate |
| Alg 3 (SLRS) | Accurate, short tokens |
| Alg 4 (TLI) | Accurate, large overlap of vocabs |

process of training a dedicated drafter. Furthermore, our approach allows practitioners to integrate SD seamlessly into existing inference pipelines without requiring any modifications to the target model's architecture or retraining procedures. The proposed algorithms expand the applicability of SD to a broader range of use cases, including models with different tokenization schemes. This is particularly relevant for practitioners and researchers who rely on pre-trained models (e.g., from the Hugging Face Hub), each with distinct vocabularies. Our methods provide practical solutions to unify heterogeneous models under a single SD framework, enhancing efficiency across diverse applications.

**Limitations.** A fundamental limitation of SD algorithms is their dependence on the acceptance rate of the drafter and the latency of its forward pass, as extensively analyzed in Timor et al. (2025). When the drafter approximates the target distribution inaccurately, the acceptance rate decreases, leading to diminished performance improvements. Our proposed methods are no exception to this constraint. Unlike standard SD that limits the drafter to in-family models, our algorithms open the door to using off-the-shelf target-drafter pairs that differ in their architecture and the way they were trained, although both can critically affect the acceptance rate. For drafters with a heterogeneous vocabulary, the inherit mismatches in token granularity might further reduce the likelihood of draft tokens being accepted. Despite these challenges, our algorithms empirically demonstrate significant accelerations not only for heterogeneous drafters (Section 5) but also homogeneous ones (Appendix D) while employing drafters that are faster than the fastest in-family model. However, for cases with insufficiently fast or accurate drafters, our methods might fail, as Appendix D shows.

## Acknowledgments

We are grateful to Roy Schwartz from The Hebrew University of Jerusalem for his valuable feedback in improving this work. We thank João Gante and the Hugging Face team for reviewing the code and providing valuable feedback that contributed to its implementation in the Transformers library.

This work was partially funded by the Israel Science Foundation (ISF grant 3698/21). Additional support was provided by a research grant to David Harel from Louis J. Lavigne and Nancy Rothman, the Carter Chapman Shreve Family Foundation, Dr. and Mrs. Donald Rivin, and the Estate of Smigel Trust.

## Impact Statement

This work lowers the cost and latency of LLM inference—making the serving of these models cheaper, faster, and more accessible to a wider range of users.

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

# A   Future Work

Future work includes assessing the effectiveness and applicability of Algorithm 3 in real-world scenarios, particularly with drafters of small vocabularies such as Wang et al., 2024, and exploring drafter adjustment strategies for Algorithm 4 to increase acceptance rates.

# B   Standard Speculative Decoding

Generating the next token via autoregressive decoding requires computing a target forward pass. Standard SD methods, like Algorithm 5, tend to utilize this target forward pass to verify multiple candidate tokens at once via a data parallelism technique known as batching, which is supported by modern hardware such as GPUs and TPUs. Running Algorithm 5 to generate the next target token requires only one target forward pass (line 6 of Algorithm 5) although the algorithm could generate between one and $i + 1$ new tokens. Since computing the target forward pass is often the slowest and most expensive operation in the inference pipeline, the ability of SD methods like Algorithm 5 to reduce the number of required target forward passes is the key to their efficiency, as was previously shown in Leviathan et al. (2023); Chen et al. (2023); Timor et al. (2025).

Algorithm 5 samples draft tokens from the drafter and then decides whether to accept or reject each draft token based on the target model's logits. The algorithm is widely used in practice and has been shown to be effective in accelerating the inference of large language models. The algorithm is lossless, meaning that it outputs tokens that distribute as the output tokens of standard autoregressive decoding.

---

**Algorithm 5** Standard Speculative Decoding (Adapted from Leviathan et al., 2023; Chen et al., 2023)

---

1: **Input:** Probability distributions $p$ and $q$ over a vocabulary $T$. Drafting lookahead $i \in \mathbb{N}$. An input prompt $c$.
2: **Output:** A sequence of tokens from $T$, containing between 1 and $i + 1$ tokens.
3: **Procedure:**
4: For $j \leftarrow 1, \ldots, i$:
5:    Sample a draft token from the drafter conditioned on the prompt and previous drafts, $d_j \sim q_{c \oplus d_1 \oplus \ldots \oplus d_{j-1}}$ (where $d_0 := c$).
6: With data parallelism (batching), compute via one target forward pass the $i + 1$ logits of the target model conditioned on the prompt and all the draft continuations, $p_c, \ p_{c \oplus d_1}, \ \cdots, \ p_{c \oplus d_1 \oplus \cdots \oplus d_i}$.
7: For $j \leftarrow 1, \ldots, i$:
8:    Let $x \leftarrow c \oplus d_1 \oplus \cdots \oplus d_{j-1}$ (where $d_0 := c$).
9:    If $p_x(d_j) \leq q_x(d_j)$, with probability $1 - \frac{p_x(d_j)}{q_x(d_j)}$, **reject** $d_j$ and go to line 11 (namely, break this for-loop).
10:    **Accept** the draft token $d_j$.
11: Let $j \in \{0, 1, \ldots, i\}$ be the number of accepted drafts. Set $x \leftarrow c \oplus d_1 \oplus \ldots \oplus d_j$.
12: Sample $t \sim r_x$ for $r_x(t) := \frac{p_x(t) - \min\{p_x(t), q_x(t)\}}{1 - \sum_{t' \in T} \min\{p_x(t'), q_x(t')\}}$ if line 9 ever rejected a token. Otherwise, sample $t \sim p_x$.
13: **Return** $d_1, \ldots, d_j, t$.

---

# C   Empirical Analysis of $\psi(t)$ Computation in Algorithm 3: Challenges and Insights

This section presents our empirical analysis of the computational complexity involved in calculating $\psi(t)$ using a real-world vocabulary. Specifically, we examine the `Qwen2-7B-Instruct` model's vocabulary to evaluate how the number of terms in $\psi(t)$ scales with the length of the target token $t$. Our findings support the theoretical prediction in Lemma 3.1, which states that the number of terms grows exponentially with the token length. We select 150,000 of the shortest tokens from a total of 151,646 tokens in the `Qwen2-7B-Instruct` vocabulary to keep the computation tractable. We then count how many ways a target token $t$ can be reconstructed by concatenating these shorter tokens. For instance, in the case of $t =$ 'hello', we found 14 valid combinations out of the 16 that would appear in a *complete* vocabulary (as defined in Section 3.4), indicating that the vocabulary of this model may be nearly complete for five-character tokens. Figure 1 lists all 14 valid combinations for the string 'hello' and visualizes them in a tree structure, where each leaf node represents a valid combination. In general, the number of forward passes of the drafter model that are required to calculate $\psi(t)$ is equal to the number of non-leaf nodes in the tree plus one. In this example, calculating $\psi($'hello'$)$ requires 16 forward passes of the drafter model, which makes Algorithm 3 with this vocabulary impractical for many target models that are considered

state-of-the-art, including the open access models StarCoder (Li et al., 2023), Llama (Dubey et al., 2024), and DeepSeek (DeepSeek-AI et al., 2025). In a similar way to the above example for the token 'hello', we decompose each of the 150,000 selected tokens into the set of all its corresponding combinations. Table 5 summarizes the statistical properties of the token

1. ['H', 'e', 'l', 'l', 'o']
2. ['H', 'e', 'l', 'lo']
3. ['H', 'e', 'll', 'o']
4. ['H', 'el', 'l', 'o']
5. ['H', 'el', 'lo']
6. ['H', 'ell', 'o']
7. ['H', 'ello']
8. ['He', 'l', 'l', 'o']
9. ['He', 'l', 'lo']
10. ['He', 'll', 'o']
11. ['Hel', 'l', 'o']
12. ['Hel', 'lo']
13. ['Hell', 'o']
14. ['Hello']

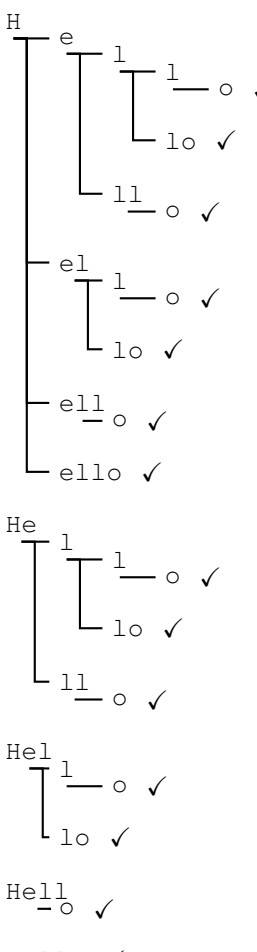

Figure 1: Left: All the 14 valid combinations of tokens from the Qwen2-7B-Instruct vocabulary that can be concatenated to form the string 'hello'. Right: Tree visualization of all these combinations. Each of the 14 checkmarks indicate a valid combination, which is a leaf in the visualized tree. In this example, calculating $\psi(t)$ from Algorithm 3 requires 16 forward passes of the drafter model, which is the number of non-leaf nodes in the tree plus one. This large number of forward passes is due to the exponential growth in the number of valid combinations as the token length increases, as shown in Figure 2.

lengths and the number of combinations for the selected tokens. The mean token length is 6.21 characters, with a standard deviation of 2.87. The mean number of combinations is 144.31, with a standard deviation of 880.98. The maximum number of combinations is 65,536. The median number of combinations is 15, and the 75th percentile is 56. Figure 2 shows the number of combinations for different token lengths. The number of combinations grows exponentially with the token length, as expected. Figure 3 shows the histogram and kernel density estimate of the number of combinations for the 150,000 selected tokens. The distribution is right-skewed, with a long tail of tokens having a large number of combinations. This exponential blow-up renders the calculation of $\psi(t)$ computationally infeasible for longer tokens, especially those among the 1,646 longest in the vocabulary. In practice, we could not even count all combinations for those tokens even after hours of computing time on a server, although only counting the combinations is an easier task than listing them. These results align with our theoretical expectations. While shorter tokens have a manageable number of decompositions, longer tokens exhibit

Table 5: Statistical summary of token length and number of combinations for a set of 150,000 shortest tokens (out of a total of 151,646 tokens) in the `Qwen2-7B-Instruct` vocabulary.

|  | Token Length (Number of Characters) | Number of Combinations |
|---|---|---|
| **Mean** | 6.21 | 144.31 |
| **Standard Deviation** | 2.87 | 880.98 |
| **Minimum** | 1.00 | 1.00 |
| **25% Percentile** | 4.00 | 7.00 |
| **50% (Median)** | 6.00 | 15.00 |
| **75% Percentile** | 8.00 | 56.00 |
| **Maximum** | 17.00 | 65536.00 |

a combinatorial explosion, underscoring the importance of using drafter models with smaller, more concise vocabularies to reduce computational overhead. Although Algorithm 3 guarantees lossless speculative decoding, the latency incurred by the computation of $\psi(t)$ may be prohibitive when the vocabulary includes very long tokens. Consequently, its applicability might be limited to models with compact or pruned vocabularies—such as MambaByte (Wang et al., 2024)—that can balance accuracy with computational feasibility. Further research should explore heuristic or approximate methods to calculate $\psi(t)$ without exhaustive enumeration. Additionally, continued work on vocabulary construction and pruning techniques that reduce redundant token entries could help mitigate these computational challenges.

## D  Speedups

We evaluate our methods on various combinations of models, tasks, and hardware setups. Tables 6 and 7 provide full benchmarks for SLEM (Algorithm 2) where the temperature is zero, and TLI (Algorithm 4) where the temperature is one, respectively. The benchmark includes widely used models: DeepSeek (DeepSeek-AI et al., 2025), Phi (Abdin et al., 2024b;a), Gemma2 (Team et al., 2024), Mixtral (Jiang et al., 2024), Qwen2.5 (Qwen et al., 2025), Vicuna (Chiang et al., 2023), Llama (Dubey et al., 2024), CodeLlama (Rozière et al., 2024), and Starcoder (Li et al., 2023). Note that for some targets, all the drafters are heterogeneous despite both target and drafter belonging to the same model family. For example, for the target `phi-4`, the drafter `Phi-3.5-mini-instruct` is heterogeneous. This is also the case for the `DeepSeek-R1-Distill-Qwen` model family, where some in-family models are heterogeneous, and therefore we cannot accelerate them using standard speculative decoding. The datasets span three tasks: code generation using HumanEval (Chen et al., 2021), text summarization using CNN-DailyMail (Nallapati et al., 2016a), and long-context task using SCROLLS (Shaham et al., 2022). For each dataset, the results are averaged over 30 prompts such that we generate between 128 and 512 new tokens for each prompt. AR denotes autoregressive decoding, SD denotes the official implementation in Hugging Face Transformers of standard speculative decoding like Algorithm 5 (Joao Gante, 2023).

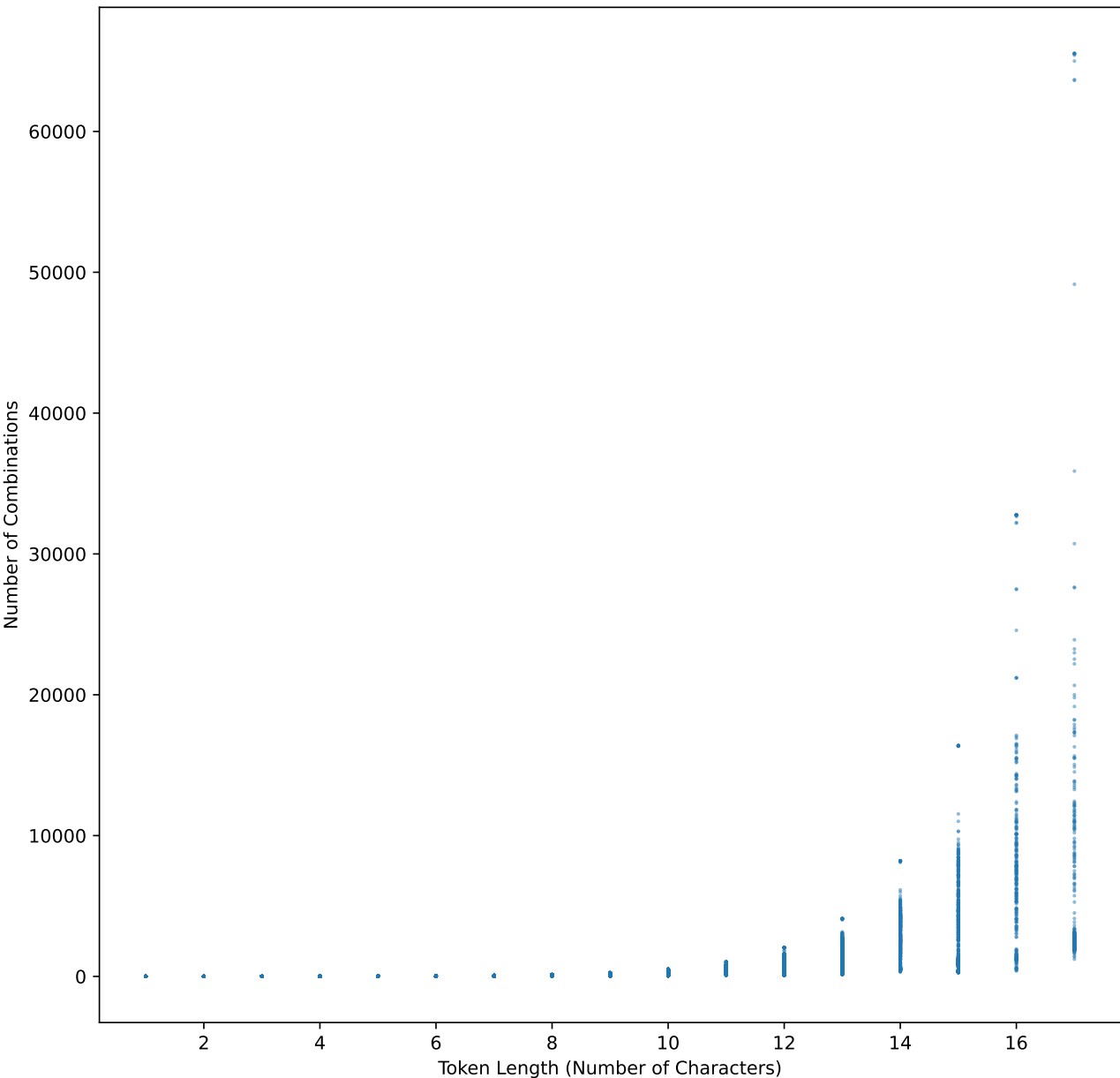

Figure 2: The number of combinations for different token lengths for the 150,000 selected tokens from the `Qwen2-7B-Instruct` vocabulary. We can see that the number of combinations grows exponentially with the token length.

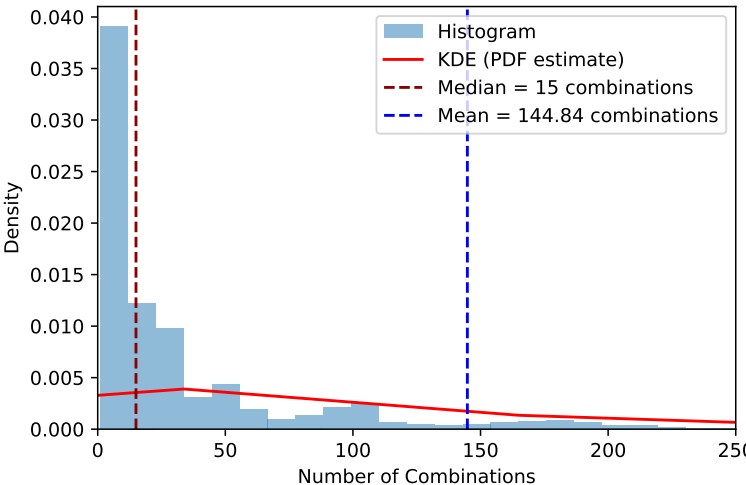

Figure 3: Histogram and Kernel Density Estimate of number of combinations for the 150,000 selected tokens from the `Qwen2-7B-Instruct` vocabulary. We can see that the number of combinations is right-skewed, with a long tail of tokens with a large number of combinations. For exact values, see Table 5.

Table 6: Full benchmark for SLEM (Algorithm 2).

| Target | Dataset | Hardware | Method | Drafter | TTFT (ms) | TPOT (ms) | Tok/s | Speedup |
|---|---|---|---|---|---|---|---|---|
| Mixtral-8x22B-Instruct-v0.1 | cnn_dailymail | 4 * H100 NVL | AR | No Drafter (Autoregressive) | **266.8** | 127.9 | 7.8 | 1.0 |
| | | | SLEM | Qwen2.5-0.5B-Instruct | 321.2 | 68.3 | 13.3 | 1.71 |
| | | | | vicuna-68m | 302.4 | **57.3** | **16.4** | **2.1** |
| | scrolls | 4 * H100 NVL | AR | No Drafter (Autoregressive) | **1331.9** | 163.0 | 6.0 | 1.0 |
| | | | SLEM | Qwen2.5-0.5B-Instruct | 1414.2 | **81.0** | **10.3** | **1.71** |
| | | | | vicuna-68m | 1344.5 | 132.5 | 7.4 | 1.24 |
| | openai_humaneval | 4 * H100 NVL | AR | No Drafter (Autoregressive) | **217.5** | 127.9 | 7.8 | 1.0 |
| | | | SLEM | Qwen2.5-0.5B-Instruct | 484.4 | **70.2** | 12.0 | 1.53 |
| | | | | vicuna-68m | 231.5 | 73.3 | **12.6** | **1.61** |
| DeepSeek-R1-Distill-Qwen-14B | scrolls | 1 * RTX 6000 | AR | No Drafter (Autoregressive) | **1481.0** | 87.5 | 10.9 | 1.0 |
| | | | SLEM | DeepSeek-R1-Distill-Qwen-1.5B | 1665.4 | 59.1 | 16.0 | 1.48 |
| | | | | vicuna-68m | 1566.8 | **56.0** | **17.3** | **1.59** |
| | cnn_dailymail | 1 * RTX 6000 | AR | No Drafter (Autoregressive) | **176.8** | 51.7 | 19.2 | 1.0 |
| | | | SLEM | DeepSeek-R1-Distill-Qwen-1.5B | 287.5 | 69.9 | 14.1 | 0.73 |
| | | | | vicuna-68m | 243.0 | **36.2** | **27.4** | **1.43** |
| | openai_humaneval | 1 * RTX 6000 | AR | No Drafter (Autoregressive) | **91.3** | 50.3 | 19.8 | 1.0 |
| | | | SLEM | tiny_starcoder_py | 113.4 | **43.8** | **22.4** | **1.14** |
| | | | | CodeLlama-7b-Instruct-hf | 256.6 | 77.5 | 12.4 | 0.63 |
| | | | | DeepSeek-R1-Distill-Qwen-1.5B | 292.5 | 70.9 | 13.6 | 0.69 |
| DeepSeek-R1-Distill-Qwen-32B | cnn_dailymail | 1 * H100 NVL | AR | No Drafter (Autoregressive) | **121.2** | 48.0 | 20.8 | 1.0 |
| | | | SLEM | DeepSeek-R1-Distill-Qwen-1.5B | 167.1 | 51.3 | 18.9 | 0.91 |
| | | | | vicuna-68m | 148.1 | **32.5** | **30.6** | **1.47** |
| | openai_humaneval | 1 * H100 NVL | AR | No Drafter (Autoregressive) | **72.0** | 48.3 | 20.7 | 1.0 |
| | | | SLEM | tiny_starcoder_py | 80.1 | **34.2** | **28.5** | **1.38** |
| | | | | CodeLlama-7b-Instruct-hf | 182.7 | 64.4 | 14.9 | 0.72 |
| | | | | DeepSeek-R1-Distill-Qwen-1.5B | 196.4 | 50.3 | 19.5 | 0.94 |
| | scrolls | 1 * H100 NVL | AR | No Drafter (Autoregressive) | **933.1** | 77.7 | 12.5 | 1.0 |
| | | | SLEM | DeepSeek-R1-Distill-Qwen-1.5B | 988.1 | **57.6** | **17.1** | **1.37** |
| | | | | vicuna-68m | 979.9 | 59.3 | 16.5 | 1.32 |
| phi-4 | scrolls | 1 * H100 NVL | AR | No Drafter (Autoregressive) | **483.9** | 47 | 21.3 | 1.0 |
| | | | SLEM | Qwen2.5-0.5B-Instruct | 457.7 | **29.5** | **33.9** | **1.59** |
| | | | | Phi-3.5-mini-instruct | 646.9 | 39.6 | 25.3 | 1.19 |
| CodeLlama-13b-Instruct-hf | humaneval | 1 * A6000 | AR | No Drafter (Autoregressive) | 70.7 | 46.8 | 21.4 | 1.0 |
| | | | SLEM | tiny_starcoder_py | **109.7** | **16.7** | **59.7** | **2.79** |
| | | | | CodeLlama-7b-Instruct-hf | 146.5 | 21.8 | 45.8 | 2.14 |
| DeepSeek-R1-Distill-Qwen-7B | cnn_dailymail | 1 * H100 NVL | AR | No Drafter (Autoregressive) | **34.7** | 19.3 | 51.8 | 1.0 |
| | | | SLEM | DeepSeek-R1-Distill-Qwen-1.5B | 85.1 | 38.6 | 24.6 | 0.48 |
| | | | | vicuna-68m | 65.2 | **17.6** | **55.2** | **1.07** |
| | openai_humaneval | 1 * H100 NVL | AR | No Drafter (Autoregressive) | **24.4** | **19.6** | **50.9** | **1.0** |
| | | | SLEM | tiny_starcoder_py | 36.1 | 23.7 | 39.6 | 0.78 |
| | | | | CodeLlama-7b-Instruct-hf | 138.0 | 54.4 | 17.5 | 0.34 |
| | | | | DeepSeek-R1-Distill-Qwen-1.5B | 149.3 | 42.2 | 22.7 | 0.45 |
| | scrolls | 1 * H100 NVL | AR | No Drafter (Autoregressive) | **221.2** | **22.4** | **44.0** | **1.0** |
| | | | SLEM | DeepSeek-R1-Distill-Qwen-1.5B | 296.2 | 41.1 | 23.5 | 0.54 |
| | | | | vicuna-68m | 245.4 | 24.6 | 39.9 | 0.91 |
| gemma-2-9b-it | scrolls | 1 * H100 NVL | AR | No Drafter (Autoregressive) | **584.0** | 95.3 | 9.9 | 1.0 |
| | | | SD | gemma-2-2b-it | 739.0 | **31.3** | **30.2** | **3.05** |
| | | | SLEM | vicuna-68m | 592.4 | 48.3 | 18.6 | 1.87 |
| | openai_humaneval | 1 * H100 NVL | AR | No Drafter (Autoregressive) | **42.6** | 37.0 | 27.0 | 1.0 |

| Target | Dataset | Hardware | Method | Drafter | TTFT (ms) | TPOT (ms) | T/s | Speedup |
|---|---|---|---|---|---|---|---|---|
| | | | SD | gemma-2-2b-it | 446.5 | 29.1 | 33.2 | 1.23 |
| | | | SLEM | vicuna-68m | 51.6 | **24.5** | **40.2** | **1.49** |
| | cnn_dailymail | 1 * H100 NVL | AR | No Drafter (Autoregressive) | **73.7** | 37.3 | 26.7 | 1.0 |
| | | | SD | gemma-2-2b-it | 125.4 | 39.7 | 24.6 | 0.92 |
| | | | SLEM | vicuna-68m | 83.9 | **26.9** | **37.1** | **1.39** |
| DeepSeek-R1-Distill-Llama-70B | openai_humaneval | 2 * A100 80GB PCIe | AR | No Drafter (Autoregressive) | 297.1 | 122.6 | 8.2 | 1.0 |
| | | | SD | CodeLlama-7b-Instruct-hf | 428.7 | 101.5 | 9.6 | 1.18 |
| | | | | DeepSeek-R1-Distill-Llama-8B | 353.5 | **54.3** | **18.3** | **2.25** |
| | | | SLEM | tiny_starcoder_py | **265.9** | | 11.8 | 1.44 |
| | | 2 * H100 NVL | AR | No Drafter (Autoregressive) | **130.1** | 76.1 | 13.1 | 1.0 |
| | | | SD | CodeLlama-7b-Instruct-hf | 277.3 | 93.3 | 10.4 | 0.79 |
| | | | | DeepSeek-R1-Distill-Llama-8B | 223.5 | **52.8** | **18.8** | **1.43** |
| | | | SLEM | DeepSeek-R1-Distill-Qwen-1.5B | 297.3 | 60.6 | 16.3 | 1.24 |
| | | | | tiny_starcoder_py | 143.3 | 63.2 | 15.6 | 1.19 |
| | cnn_dailymail | 2 * H100 NVL | AR | No Drafter (Autoregressive) | **230.7** | 77.6 | 12.9 | 1.0 |
| | | | SD | DeepSeek-R1-Distill-Llama-8B | 452.5 | 78.7 | 12.5 | 0.97 |
| | | | | Llama-3.1-8B | 277.9 | 74.4 | 13.4 | 1.04 |
| | | | | Llama-3.2-1B | 242.3 | **51.4** | **19.4** | **1.51** |
| | | | | Llama-3.2-3B | 252.1 | 66.8 | 14.9 | 1.16 |
| | | | SLEM | DeepSeek-R1-Distill-Qwen-1.5B | 296.1 | 72.3 | 13.6 | 1.06 |
| | | | | vicuna-68m | 263.8 | 51.5 | 19.3 | 1.5 |
| | scrolls | 2 * H100 NVL | AR | No Drafter (Autoregressive) | **1836.9** | 127.1 | 7.7 | 1.0 |
| | | | SD | DeepSeek-R1-Distill-Llama-8B | 2121.4 | 88.0 | 10.9 | 1.42 |
| | | | SLEM | DeepSeek-R1-Distill-Qwen-1.5B | 1890.9 | **85.8** | **11.3** | **1.47** |
| DeepSeek-R1-Distill-Llama-8B | scrolls | 1 * H100 NVL | AR | No Drafter (Autoregressive) | **245.3** | 34.3 | 27.9 | 1.0 |
| | | | SD | Llama-3.2-1B | 283.1 | **24.6** | **39.2** | **1.41** |
| | | | | Llama-3.2-3B | 353.1 | 35.2 | 27.3 | 0.98 |
| | | | SLEM | DeepSeek-R1-Distill-Qwen-1.5B | 315.7 | 45.2 | 20.2 | 0.73 |
| | | | | vicuna-68m | 263.4 | 27.7 | 34.9 | 1.25 |
| | cnn_dailymail | 1 * H100 NVL | AR | No Drafter (Autoregressive) | **38.9** | 21.8 | 45.9 | 1.0 |
| | | | SD | Llama-3.2-1B | 48.2 | 25.6 | 38.6 | 0.84 |
| | | | | Llama-3.2-3B | 57.2 | 38.1 | 26.0 | 0.57 |
| | | | SLEM | DeepSeek-R1-Distill-Qwen-1.5B | 88.2 | 43.2 | 22.7 | 0.5 |
| | | | | vicuna-68m | 66.9 | **19.7** | **50.0** | **1.09** |
| | openai_humaneval | 1 * H100 NVL | AR | No Drafter (Autoregressive) | **31.9** | **21.8** | **45.8** | **1.0** |
| | | | SD | CodeLlama-7b-Instruct-hf | 144.0 | 72.3 | 12.5 | 0.27 |
| | | | SLEM | tiny_starcoder_py | 36.7 | 37.1 | 25.8 | 0.56 |
| | | 1 * RTX 6000 | AR | No Drafter (Autoregressive) | **73.4** | 40.8 | 24.5 | **1.0** |
| | | | SD | CodeLlama-7b-Instruct-hf | 279.8 | 120.1 | 7.9 | 0.32 |
| | | | SLEM | tiny_starcoder_py | 96.4 | 52.6 | 18.2 | 0.74 |
| | | | | DeepSeek-R1-Distill-Qwen-1.5B | 246.2 | 42.7 | 20.4 | 0.83 |

Table 7: Full benchmark for TLI (Algorithm 4).

| Target | Dataset | Hardware | Method | Drafter | TTFT (ms) | TPOT (ms) | Tok/s | Speedup |
|---|---|---|---|---|---|---|---|---|
| Mixtral-8x22B-Instruct-v0.1 | scrolls | 4 * H100 NVL | AR | No Drafter (Autoregressive) | 1334.7 | 168.7 | 5.9 | 1.0 |
| | | | TLI | Qwen2.5-0.5B-Instruct | 1372.6 | **97.8** | 9.9 | **1.69** |
| | | | | vicuna-68m | **1329.7** | 138.2 | 7.2 | 1.22 |
| | openai_humaneval | 4 * H100 NVL | AR | No Drafter (Autoregressive) | **217.5** | 128.1 | 7.8 | 1.0 |
| | | | TLI | Qwen2.5-0.5B-Instruct | 266.9 | 90.6 | 10.9 | 1.4 |
| | | | | vicuna-68m | 228.5 | **74.8** | **13.0** | **1.67** |
| | cnn_dailymail | 4 * H100 NVL | AR | No Drafter (Autoregressive) | **266.8** | 128.1 | 7.8 | 1.0 |
| | | | TLI | Qwen2.5-0.5B-Instruct | 294.5 | 88.9 | 11.2 | 1.43 |
| | | | | vicuna-68m | 297.3 | **81.0** | **11.9** | **1.53** |
| phi-4 | scrolls | 1 * H100 NVL | AR | No Drafter (Autoregressive) | 487.4 | 47.2 | 21.2 | 1.0 |
| | | | TLI | Qwen2.5-0.5B-Instruct | **454.7** | **32.5** | **30.8** | **1.45** |
| | | | | Phi-3.5-mini-instruct | 610.4 | 46.0 | 21.7 | 1.03 |
| CodeLlama-13b-Instruct-hf | humaneval | 1 * A6000 | AR | No Drafter (Autoregressive) | 70.5 | 45.3 | 22.1 | 1.0 |
| | | | TLI | tiny_starcoder_py | **65.1** | **25.9** | **38.5** | 1.74 |
| | | | | CodeLlama-7b-Instruct-hf | 141.3 | 25.6 | 39.1 | 1.77 |
| DeepSeek-R1-Distill-Qwen-14B | scrolls | 1 * RTX 6000 | AR | No Drafter (Autoregressive) | **1479.5** | 88.3 | 10.8 | 1.0 |
| | | | TLI | DeepSeek-R1-Distill-Qwen-1.5B | 1640.7 | 61.6 | 16.1 | 1.5 |
| | | | | vicuna-68m | 1502.2 | **57.2** | **17.1** | **1.59** |
| | cnn_dailymail | 1 * RTX 6000 | AR | No Drafter (Autoregressive) | **176.1** | 54.4 | 18.4 | 1.0 |
| | | | TLI | DeepSeek-R1-Distill-Qwen-1.5B | 240.5 | 44.7 | 21.4 | 1.16 |
| | | | | vicuna-68m | 202.4 | **40.6** | **24.1** | **1.31** |
| | openai_humaneval | 1 * RTX 6000 | AR | No Drafter (Autoregressive) | **90.4** | 50.9 | 19.6 | 1.0 |
| | | | TLI | tiny_starcoder_py | 93.9 | **38.6** | **25.4** | **1.3** |
| | | | | CodeLlama-7b-Instruct-hf | 150.2 | 66.0 | 14.6 | 0.75 |
| | | | | DeepSeek-R1-Distill-Qwen-1.5B | 172.6 | 45.6 | 21.2 | 1.08 |
| DeepSeek-R1-Distill-Qwen-7B | cnn_dailymail | 1 * H100 NVL | AR | No Drafter (Autoregressive) | **35.0** | 19.8 | 50.6 | 1.0 |
| | | | TLI | DeepSeek-R1-Distill-Qwen-1.5B | 95.0 | 27.2 | 36.6 | 0.72 |
| | | | | vicuna-68m | 108.5 | **18.4** | **54.2** | **1.07** |
| | openai_humaneval | 1 * H100 NVL | AR | No Drafter (Autoregressive) | **23.4** | 20.0 | 49.9 | **1.0** |
| | | | TLI | tiny_starcoder_py | 40.0 | 22.3 | 44.7 | 0.9 |
| | | | | CodeLlama-7b-Instruct-hf | 59.6 | 39.2 | 25.3 | 0.51 |
| | | | | DeepSeek-R1-Distill-Qwen-1.5B | 88.3 | 24.3 | 40.9 | 0.82 |
| | scrolls | 1 * H100 NVL | AR | No Drafter (Autoregressive) | **220.5** | **22.8** | **43.2** | **1.0** |
| | | | TLI | DeepSeek-R1-Distill-Qwen-1.5B | 296.9 | 29.1 | 34.2 | 0.79 |
| | | | | vicuna-68m | 238.6 | 25.0 | 39.2 | 0.91 |
| gemma-2-9b-it | scrolls | 1 * H100 NVL | AR | No Drafter (Autoregressive) | **585.1** | 90.6 | 10.4 | 1.0 |

| Target | Dataset | Hardware | Method | Drafter | TTFT (ms) | TPOT (ms) | T/s | Speedup |
|---|---|---|---|---|---|---|---|---|
| | | | TLI | vicuna-68m | 603.0 | 46.0 | 21.4 | 2.04 |
| | | | SD | gemma-2-2b-it | 742.3 | **37.7** | **26.0** | **2.49** |
| | openai_humaneval | 1 * H100 NVL | AR | No Drafter (Autoregressive) | **42.6** | 37.3 | 26.8 | 1.0 |
| | | | TLI | vicuna-68m | 92.8 | **25.1** | **39.2** | **1.46** |
| | | | SD | gemma-2-2b-it | 384.1 | 27.2 | 36.4 | 1.36 |
| | cnn_dailymail | 1 * H100 NVL | AR | No Drafter (Autoregressive) | **73.7** | 37.8 | 26.5 | 1.0 |
| | | | TLI | vicuna-68m | 100.1 | **30.0** | **33.2** | **1.26** |
| | | | SD | gemma-2-2b-it | 117.8 | 33.5 | 29.8 | 1.13 |
| DeepSeek-R1-Distill-Llama-70B | openai_humaneval | 2 * A100 80GB PCIe | AR | No Drafter (Autoregressive) | **244.0** | 123.5 | 8.1 | 1.0 |
| | | | TLI | tiny_starcoder_py | 258.7 | 85.5 | 11.7 | 1.44 |
| | | | SD | CodeLlama-7b-Instruct-hf | 317.3 | 87.2 | 11.4 | 1.41 |
| | | | | DeepSeek-R1-Distill-Llama-8B | 358.2 | **53.1** | **18.6** | **2.3** |
| | | 2 * H100 NVL | AR | No Drafter (Autoregressive) | **129.9** | 76.7 | 13.0 | 1.0 |
| | | | TLI | tiny_starcoder_py | 147.9 | 55.3 | 18.0 | 1.38 |
| | | | SD | CodeLlama-7b-Instruct-hf | 214.5 | 68.8 | 14.5 | 1.11 |
| | | | | DeepSeek-R1-Distill-Llama-8B | 179.7 | **44.6** | **22.4** | **1.72** |
| | | | TLI | DeepSeek-R1-Distill-Qwen-1.5B | 220.4 | 45.5 | 21.9 | 1.68 |
| | scrolls | 2 * H100 NVL | AR | No Drafter (Autoregressive) | **1837.1** | 126.6 | 7.7 | 1.0 |
| | | | SD | DeepSeek-R1-Distill-Llama-8B | 2059.4 | **65.1** | **15.2** | **1.98** |
| | | | TLI | DeepSeek-R1-Distill-Qwen-1.5B | 1898.5 | 70.9 | 13.9 | 1.82 |
| | cnn_dailymail | 2 * H100 NVL | AR | No Drafter (Autoregressive) | **231.2** | 77.9 | 12.8 | 1.0 |
| | | | SD | DeepSeek-R1-Distill-Llama-8B | 342.5 | 58.2 | 17.1 | 1.33 |
| | | | TLI | DeepSeek-R1-Distill-Qwen-1.5B | 315.5 | 59.7 | 16.7 | 1.3 |
| | | | | vicuna-68m | 263.4 | 55.8 | 17.8 | 1.39 |
| | | | SD | Llama-3.1-8B | 262.3 | 59.6 | 16.7 | 1.31 |
| | | | | Llama-3.2-1B | 248.5 | **51.3** | **19.4** | **1.51** |
| | | | | Llama-3.2-3B | 259.6 | 56.7 | 17.5 | 1.37 |
| DeepSeek-R1-Distill-Qwen-32B | scrolls | 1 * H100 NVL | AR | No Drafter (Autoregressive) | **946.4** | 77.3 | 12.5 | 1.0 |
| | | | TLI | DeepSeek-R1-Distill-Qwen-1.5B | 997.8 | **43.0** | **22.7** | **1.82** |
| | | | | vicuna-68m | 977.1 | 61.6 | 15.9 | 1.27 |
| | openai_humaneval | 1 * H100 NVL | AR | No Drafter (Autoregressive) | **72.2** | 48.6 | 20.6 | 1.0 |
| | | | TLI | tiny_starcoder_py | 86.2 | 35.5 | 28.1 | 1.37 |
| | | | | CodeLlama-7b-Instruct-hf | 123.4 | 49.5 | 20.1 | 0.97 |
| | | | | DeepSeek-R1-Distill-Qwen-1.5B | 147.4 | **28.5** | **35.0** | **1.7** |
| | cnn_dailymail | 1 * H100 NVL | AR | No Drafter (Autoregressive) | **121.5** | 49.0 | 20.4 | 1.0 |
| | | | TLI | DeepSeek-R1-Distill-Qwen-1.5B | 167.5 | 37.9 | 26.1 | 1.28 |
| | | | | vicuna-68m | 146.4 | **34.2** | **29.1** | **1.42** |
| DeepSeek-R1-Distill-Llama-8B | scrolls | 1 * H100 NVL | AR | No Drafter (Autoregressive) | **246.7** | 38.6 | 24.9 | 1.0 |
| | | | TLI | DeepSeek-R1-Distill-Qwen-1.5B | 324.4 | 33.5 | 29.6 | 1.19 |
| | | | | vicuna-68m | 256.5 | 28.1 | 34.5 | 1.39 |
| | | | SD | Llama-3.2-1B | 295.2 | **24.9** | **39.7** | **1.6** |
| | | | | Llama-3.2-3B | 355.9 | 31.7 | 31.2 | 1.25 |
| | cnn_dailymail | 1 * H100 NVL | AR | No Drafter (Autoregressive) | **39.6** | 22.3 | 44.9 | 1.0 |
| | | | TLI | DeepSeek-R1-Distill-Qwen-1.5B | 93.4 | 31.9 | 31.1 | 0.69 |
| | | | | vicuna-68m | 75.4 | **20.3** | **49.1** | **1.09** |
| | | | SD | Llama-3.2-1B | 51.8 | 22.6 | 44.2 | 0.98 |
| | | | | Llama-3.2-3B | 60.2 | 29.2 | 34.2 | 0.76 |
| | openai_humaneval | 1 * H100 NVL | AR | No Drafter (Autoregressive) | **31.2** | 22.3 | 44.8 | 1.0 |
| | | | TLI | tiny_starcoder_py | 43.5 | 23.6 | 42.1 | 0.94 |
| | | | SD | CodeLlama-7b-Instruct-hf | 99.0 | 38.0 | 26.0 | 0.58 |
| | | 1 * RTX 6000 | AR | No Drafter (Autoregressive) | **73.4** | 41.1 | 24.3 | 1.0 |
| | | | TLI | tiny_starcoder_py | 82.5 | 39.5 | 25.1 | 1.03 |
| | | | | CodeLlama-7b-Instruct-hf | 218.6 | 63.0 | 15.7 | 0.65 |
| | | | | DeepSeek-R1-Distill-Qwen-1.5B | 145.8 | **35.7** | **26.0** | **1.07** |

# E   Vocabularies and Overlap

This section examines the vocabularies of widely used off-the-shelf target and drafter models. Table 8 shows the vocabulary sizes of widely used target and drafter models. Table 9 shows the vocabulary overlap between the target and drafter models. Table 10 shows the ratio of the number of tokens in the intersection between the target and draft vocabularies $|T' \cap D'|$ to the number of tokens in the target vocabulary $|T'|$, considering only the tokens that appeared in 50 randomly selected prompts for the given task.

Table 8: Vocabulary sizes of widely used target and drafter models.

| Target Model | Vocabulary Size $|T|$ |
|---|---|
| google/Gemma-2-9b | 256,000 |
| meta-llama/Llama-3.1-70B | 128,256 |
| mistralai/Mixtral-8x22B-Instruct-v0.1 | 32,768 |
| microsoft/Phi-3-medium-128k-instruct | 32,011 |
| codellama/CodeLlama-13b-Instruct-hf | 32,016 |
| **Drafter Model** | **Vocabulary Size $|D|$** |
| Qwen/Qwen2-0.5B-Instruct | 151,646 |
| bigcode/tiny_starcoder_py | 49,152 |
| double7/vicuna-68m | 32,000 |

Table 9: Vocabulary overlap metrics for widely used target and drafter models: the size of the intersection between the target vocabulary and the draft vocabulary, and the ratio of the intersection size to the target vocabulary size. We can see a wide range of overlap sizes and ratios.

| Target Model | Drafter Model | $|T \cap D|$ | $|T \cap D|/|T|$ |
|---|---|---|---|
| Llama-3.1-70B | Qwen2-0.5B-Instruct | 109,566 | 0.85 |
| Gemma-2-9b | vicuna-68m | 30,489 | 0.12 |
| Mixtral-8x22B-Instruct-v0.1 | vicuna-68m | 24,184 | 0.74 |
| Mixtral-8x22B-Instruct-v0.1 | Qwen2-0.5B-Instruct | 10,566 | 0.32 |
| Phi-3-medium-128k-instruct | Qwen2-0.5B-Instruct | 9,588 | 0.30 |
| CodeLlama-13b-Instruct-hf | tiny_starcoder_py | 8,481 | 0.26 |

# F   Injectivity of Tokenizers Under the CMM-DM Dataset

The experiment sampled uniformly at random examples from the CNN-DM dataset (Nallapati et al., 2016b), and took the prefix of 100 characters from each example. Using a SentencePiece tokenizer (Kudo & Richardson, 2018) or various other Hugging Face Transformers tokenizers (Wolf et al., 2020), we encoded the prefix into tokens, and then decoded the tokens back into text. We then checked whether the original prefix could be recovered by checking whether $s = \text{decode}(\text{encode}(s))$. While a tokenizer may implement a non-injective function in general, this experiment specifically tested its injectivity on the given dataset. The results of our experiment are summarized in Table 11.

# G   Proofs

**Theorem 3.1.** *Let $p$ be a non-trivial target probability distribution over a vocabulary $T$, where there exist $t_1, t_2 \in T$ such that $p(t_1) \neq p(t_2)$. Let $q$ be the drafter probability distribution over the same vocabulary $T$. If $q = p$, namely, the drafter is another instance of the target model, then the expected acceptance rate of the exact matching method $\alpha_{EM}$ is strictly less than the expected acceptance rate of the standard speculative decoding method $\alpha_{SD}$. Namely, it holds that $\alpha_{EM} < \alpha_{SD}$.*

*Proof.* The expected acceptance rate of the standard speculative decoding verification method is $\alpha_{\text{SD}} = \sum_{t \in T} \min\{p(t), q(t)\}$ by Leviathan et al. (2023). If $q = p$, we have $\alpha_{\text{SD}} = \sum_{t \in T} \min\{p(t), p(t)\} = \sum_{t \in T} p(t) = 1$. For exact matching, a token $t$ is accepted if it is sampled by both the draft and the target models. Since these are independent events, the probability of accepting $t$ is $p(t) \cdot p(t) = p(t)^2$. Thus, we have $\alpha_{\text{EM}} = \sum_{t \in T} p(t)^2$. For any $p(t)$ such that $0 < p(t) < 1$, it holds that $p(t)^2 < p(t)$. Summing over all tokens $t \in T$, we get that $\sum_{t \in T} p(t)^2 < \sum_{t \in T} p(t) = 1$. Therefore, $\alpha_{\text{EM}} < \alpha_{\text{SD}}$ for any non-trivial target distribution $p$. □

**Theorem 3.2.** *For any token in the target vocabulary $t \in T$, Algorithm 3 outputs the token $t$ with probability $p(t)$ if we define $\psi(t) := \sum\limits_{d_1, d_2, \ldots, d_i \ : \ t=T(d_1 \oplus \ldots \oplus d_i)_1} \prod\limits_{j \in \{1, \ldots, i\}} q(d_j)$. Namely, Algorithm 3 is lossless.*

*Proof.* Denote the probability of accepting the token $t_1$ by $\Pr[\text{accept } t \mid t]$. We have that $\Pr[\text{accept } t \mid t] = 1$ if $p(t) \geq \psi(t)$, and $\frac{p(t)}{\psi(t)}$ otherwise. We also have that the probability of sampling tokens from $q$ such that their concatenation forms $t$ is $\psi(t)$. Therefore, $\sum_t \Pr[\text{accept } t] = \sum_t \Pr[\text{accept } t \mid t] \cdot \Pr[t] = \sum_t \min\{p(t), \psi(t)\}$. The probability of outputting $t$ is then $\Pr[\text{output } t] = \Pr[\text{accept } t] + (1 - \sum_t \Pr[\text{accept } t]) \cdot \frac{p(t) - \min\{p(t), \psi(t)\}}{1 - \sum_{t'} \min\{p(t'), \psi(t')\}} = p(t)$. □

**Lemma 3.1.** *For a target token $t$ of length $m \leq n$ in a complete vocabulary $D_n$ that contains all possible strings of length up to $n$ over a fixed alphabet $\Sigma$, the number of distinct sequences of draft tokens $d_1, \ldots, d_i$ such that their concatenation $d_1 \oplus \ldots \oplus d_i$ starts with $t$, namely, $T(d_1 \oplus \ldots \oplus d_i)_1 = t$, is $2^{m-1}$.*

*Proof.* We can approach this counting problem by considering it as a combinatorial composition, specifically the number of ways to write the length $m$ of the target token $t$ as the sum of a sequence of strictly positive integers. Consider the token $t$ of length $m$, which can be decomposed into a sequence of tokens $t_1, t_2, \ldots, t_m$. Each possible partition of $m$ into smaller segments corresponds to a unique way of concatenating draft tokens from the vocabulary. The problem can be reduced to counting how many distinct ways we can concatenate these tokens to obtain the desired target token $t$. There are exactly $2^{m-1}$ ways to achieve this because, at each position between the tokens, we have two choices: either to concatenate the next token with the previous segment or to keep it separate. For example, given the sequence $t_1, t_2, \ldots, t_m$, the possible compositions include $(t_1 \oplus t_2), t_3, \ldots, t_m$; $t_1, (t_2 \oplus t_3), \ldots, t_m$; and $(t_1 \oplus t_2 \oplus t_3), t_4, \ldots, t_m$, and so forth, covering all

Table 10: The ratio of the number of tokens in the intersection between the target and draft vocabularies $|T' \cap D'|$ to the number of tokens in the target vocabulary $|T'|$, considering only the tokens that appeared in 50 randomly selected prompts for the given task. Note that $|T' \cap D'|/|T'|$ for a given task could differ from $|T \cap D|/|T|$ because some tokens of $T$ or $D$ might not appear in the prompts of the given task.

| Target Model | Drafter Model | Task | Dataset | $\frac{|T' \cap D'|}{|T'|}$ |
|---|---|---|---|---|
| CodeLlama-13b-Instruct-hf | CodeLlama-7b-Instruct-hf | coding | openai_humaneval | 1.0 |
| CodeLlama-13b-Instruct-hf | tiny_starcoder_py | coding | openai_humaneval | 0.86 |
| DeepSeek-R1-Distill-Llama-70B | CodeLlama-7b-Instruct-hf | coding | openai_humaneval | 0.84 |
| DeepSeek-R1-Distill-Llama-70B | DeepSeek-R1-Distill-Llama-8B | coding | openai_humaneval | 1.0 |
| DeepSeek-R1-Distill-Llama-70B | DeepSeek-R1-Distill-Llama-8B | long-ctx summ | scrolls | 1.0 |
| DeepSeek-R1-Distill-Llama-70B | DeepSeek-R1-Distill-Llama-8B | summ | cnn_dailymail | 1.0 |
| DeepSeek-R1-Distill-Llama-70B | DeepSeek-R1-Distill-Qwen-1.5B | coding | openai_humaneval | 1.0 |
| DeepSeek-R1-Distill-Llama-70B | DeepSeek-R1-Distill-Qwen-1.5B | long-ctx summ | scrolls | 1.0 |
| DeepSeek-R1-Distill-Llama-70B | DeepSeek-R1-Distill-Qwen-1.5B | summ | cnn_dailymail | 1.0 |
| DeepSeek-R1-Distill-Llama-70B | Llama-3.1-8B | long-ctx summ | scrolls | 1.0 |
| DeepSeek-R1-Distill-Llama-70B | Llama-3.1-8B | summ | cnn_dailymail | 1.0 |
| DeepSeek-R1-Distill-Llama-70B | Llama-3.2-1B | long-ctx summ | scrolls | 1.0 |
| DeepSeek-R1-Distill-Llama-70B | Llama-3.2-1B | summ | cnn_dailymail | 1.0 |
| DeepSeek-R1-Distill-Llama-70B | Llama-3.2-3B | long-ctx summ | scrolls | 1.0 |
| DeepSeek-R1-Distill-Llama-70B | Llama-3.2-3B | summ | cnn_dailymail | 1.0 |
| DeepSeek-R1-Distill-Llama-70B | tiny_starcoder_py | coding | openai_humaneval | 0.94 |
| DeepSeek-R1-Distill-Llama-70B | vicuna-68m | long-ctx summ | scrolls | 0.97 |
| DeepSeek-R1-Distill-Llama-70B | vicuna-68m | summ | cnn_dailymail | 0.98 |
| DeepSeek-R1-Distill-Llama-8B | CodeLlama-7b-Instruct-hf | coding | openai_humaneval | 0.77 |
| DeepSeek-R1-Distill-Llama-8B | DeepSeek-R1-Distill-Qwen-1.5B | coding | openai_humaneval | 1.0 |
| DeepSeek-R1-Distill-Llama-8B | DeepSeek-R1-Distill-Qwen-1.5B | long-ctx summ | scrolls | 1.0 |
| DeepSeek-R1-Distill-Llama-8B | DeepSeek-R1-Distill-Qwen-1.5B | summ | cnn_dailymail | 1.0 |
| DeepSeek-R1-Distill-Llama-8B | Llama-3.2-1B | long-ctx summ | scrolls | 1.0 |
| DeepSeek-R1-Distill-Llama-8B | Llama-3.2-1B | summ | cnn_dailymail | 1.0 |
| DeepSeek-R1-Distill-Llama-8B | Llama-3.2-3B | long-ctx summ | scrolls | 1.0 |
| DeepSeek-R1-Distill-Llama-8B | Llama-3.2-3B | summ | cnn_dailymail | 1.0 |
| DeepSeek-R1-Distill-Llama-8B | tiny_starcoder_py | coding | openai_humaneval | 0.94 |
| DeepSeek-R1-Distill-Llama-8B | vicuna-68m | long-ctx summ | scrolls | 0.98 |
| DeepSeek-R1-Distill-Llama-8B | vicuna-68m | summ | cnn_dailymail | 0.98 |
| DeepSeek-R1-Distill-Qwen-14B | CodeLlama-7b-Instruct-hf | coding | openai_humaneval | 0.83 |
| DeepSeek-R1-Distill-Qwen-14B | DeepSeek-R1-Distill-Qwen-1.5B | coding | openai_humaneval | 1.0 |
| DeepSeek-R1-Distill-Qwen-14B | DeepSeek-R1-Distill-Qwen-1.5B | long-ctx summ | scrolls | 1.0 |
| DeepSeek-R1-Distill-Qwen-14B | DeepSeek-R1-Distill-Qwen-1.5B | summ | cnn_dailymail | 1.0 |
| DeepSeek-R1-Distill-Qwen-14B | tiny_starcoder_py | coding | openai_humaneval | 0.93 |
| DeepSeek-R1-Distill-Qwen-14B | vicuna-68m | long-ctx summ | scrolls | 0.98 |
| DeepSeek-R1-Distill-Qwen-14B | vicuna-68m | summ | cnn_dailymail | 0.99 |
| DeepSeek-R1-Distill-Qwen-32B | CodeLlama-7b-Instruct-hf | coding | openai_humaneval | 0.83 |
| DeepSeek-R1-Distill-Qwen-32B | DeepSeek-R1-Distill-Qwen-1.5B | coding | openai_humaneval | 1.0 |
| DeepSeek-R1-Distill-Qwen-32B | DeepSeek-R1-Distill-Qwen-1.5B | long-ctx summ | scrolls | 1.0 |
| DeepSeek-R1-Distill-Qwen-32B | DeepSeek-R1-Distill-Qwen-1.5B | summ | cnn_dailymail | 1.0 |
| DeepSeek-R1-Distill-Qwen-32B | tiny_starcoder_py | coding | openai_humaneval | 0.93 |
| DeepSeek-R1-Distill-Qwen-32B | vicuna-68m | long-ctx summ | scrolls | 0.98 |
| DeepSeek-R1-Distill-Qwen-32B | vicuna-68m | summ | cnn_dailymail | 0.98 |
| DeepSeek-R1-Distill-Qwen-7B | CodeLlama-7b-Instruct-hf | coding | openai_humaneval | 0.83 |
| DeepSeek-R1-Distill-Qwen-7B | DeepSeek-R1-Distill-Qwen-1.5B | coding | openai_humaneval | 1.0 |
| DeepSeek-R1-Distill-Qwen-7B | DeepSeek-R1-Distill-Qwen-1.5B | long-ctx summ | scrolls | 1.0 |
| DeepSeek-R1-Distill-Qwen-7B | DeepSeek-R1-Distill-Qwen-1.5B | summ | cnn_dailymail | 1.0 |
| DeepSeek-R1-Distill-Qwen-7B | tiny_starcoder_py | coding | openai_humaneval | 0.93 |
| DeepSeek-R1-Distill-Qwen-7B | vicuna-68m | long-ctx summ | scrolls | 0.98 |
| DeepSeek-R1-Distill-Qwen-7B | vicuna-68m | summ | cnn_dailymail | 0.99 |
| Llama-3.1-70B | Llama-3.1-8B | coding | openai_humaneval | 1.0 |
| Llama-3.1-70B | Llama-3.1-8B | long-ctx summ | scrolls | 1.0 |
| Llama-3.1-70B | Llama-3.1-8B | summ | cnn_dailymail | 1.0 |
| Llama-3.1-70B | Llama-3.2-1B | coding | openai_humaneval | 1.0 |
| Llama-3.1-70B | Llama-3.2-1B | long-ctx summ | scrolls | 1.0 |
| Llama-3.1-70B | Llama-3.2-1B | summ | cnn_dailymail | 1.0 |
| Llama-3.1-70B | Llama-3.2-3B | coding | openai_humaneval | 1.0 |
| Llama-3.1-70B | Llama-3.2-3B | long-ctx summ | scrolls | 1.0 |
| Llama-3.1-70B | Llama-3.2-3B | summ | cnn_dailymail | 1.0 |
| Llama-3.1-70B | Qwen2.5-0.5B-Instruct | coding | openai_humaneval | 1.0 |
| Llama-3.1-70B | Qwen2.5-0.5B-Instruct | long-ctx summ | scrolls | 1.0 |
| Llama-3.1-70B | Qwen2.5-0.5B-Instruct | summ | cnn_dailymail | 1.0 |
| Llama-3.1-70B-Instruct | Llama-3.1-8B-Instruct | coding | openai_humaneval | 1.0 |
| Llama-3.1-70B-Instruct | Llama-3.1-8B-Instruct | long-ctx summ | scrolls | 1.0 |
| Llama-3.1-70B-Instruct | Llama-3.1-8B-Instruct | summ | cnn_dailymail | 1.0 |
| Llama-3.1-70B-Instruct | Llama-3.2-1B-Instruct | coding | openai_humaneval | 1.0 |
| Llama-3.1-70B-Instruct | Llama-3.2-1B-Instruct | long-ctx summ | scrolls | 1.0 |
| Llama-3.1-70B-Instruct | Llama-3.2-1B-Instruct | summ | cnn_dailymail | 1.0 |
| Llama-3.1-70B-Instruct | Llama-3.2-3B-Instruct | coding | openai_humaneval | 1.0 |
| Llama-3.1-70B-Instruct | Llama-3.2-3B-Instruct | long-ctx summ | scrolls | 1.0 |
| Llama-3.1-70B-Instruct | Llama-3.2-3B-Instruct | summ | cnn_dailymail | 1.0 |
| Llama-3.1-70B-Instruct | Qwen2.5-0.5B-Instruct | coding | openai_humaneval | 1.0 |
| Llama-3.1-70B-Instruct | Qwen2.5-0.5B-Instruct | long-ctx summ | scrolls | 1.0 |
| Llama-3.1-70B-Instruct | Qwen2.5-0.5B-Instruct | summ | cnn_dailymail | 1.0 |
| Llama-3.1-70B-Instruct | vicuna-68m | coding | openai_humaneval | 0.84 |
| Llama-3.1-70B-Instruct | vicuna-68m | long-ctx summ | scrolls | 0.97 |
| Llama-3.1-70B-Instruct | vicuna-68m | summ | cnn_dailymail | 0.99 |
| Llama-3.1-8B-Instruct | Llama-3.2-1B-Instruct | coding | openai_humaneval | 1.0 |
| Llama-3.1-8B-Instruct | Llama-3.2-1B-Instruct | long-ctx summ | scrolls | 1.0 |
| Llama-3.1-8B-Instruct | Llama-3.2-1B-Instruct | summ | cnn_dailymail | 1.0 |
| Llama-3.1-8B-Instruct | Llama-3.2-3B-Instruct | coding | openai_humaneval | 1.0 |
| Llama-3.1-8B-Instruct | Llama-3.2-3B-Instruct | long-ctx summ | scrolls | 1.0 |
| Llama-3.1-8B-Instruct | Llama-3.2-3B-Instruct | summ | cnn_dailymail | 1.0 |
| Llama-3.1-8B-Instruct | Qwen2.5-0.5B-Instruct | coding | openai_humaneval | 1.0 |
| Llama-3.1-8B-Instruct | Qwen2.5-0.5B-Instruct | long-ctx summ | scrolls | 1.0 |
| Llama-3.1-8B-Instruct | Qwen2.5-0.5B-Instruct | summ | cnn_dailymail | 1.0 |
| Llama-3.1-8B-Instruct | vicuna-68m | coding | openai_humaneval | 0.84 |
| Llama-3.1-8B-Instruct | vicuna-68m | long-ctx summ | scrolls | 0.98 |
| Llama-3.1-8B-Instruct | vicuna-68m | summ | cnn_dailymail | 0.98 |
| Mixtral-8x22B-Instruct-v0.1 | Qwen2.5-0.5B-Instruct | coding | openai_humaneval | 0.78 |
| Mixtral-8x22B-Instruct-v0.1 | Qwen2.5-0.5B-Instruct | long-ctx summ | scrolls | 0.89 |
| Mixtral-8x22B-Instruct-v0.1 | Qwen2.5-0.5B-Instruct | summ | cnn_dailymail | 0.9 |
| Mixtral-8x22B-Instruct-v0.1 | vicuna-68m | coding | openai_humaneval | 0.99 |
| Mixtral-8x22B-Instruct-v0.1 | vicuna-68m | long-ctx summ | scrolls | 0.99 |
| Mixtral-8x22B-Instruct-v0.1 | vicuna-68m | summ | cnn_dailymail | 0.99 |
| Qwen2.5-1.5B-Instruct | Qwen2.5-0.5B-Instruct | long-ctx summ | scrolls | 1.0 |
| Qwen2.5-1.5B-Instruct | vicuna-68m | long-ctx summ | scrolls | 0.98 |
| gemma-2-9b-it | gemma-2-2b-it | coding | openai_humaneval | 1.0 |
| gemma-2-9b-it | gemma-2-2b-it | long-ctx summ | scrolls | 1.0 |
| gemma-2-9b-it | gemma-2-2b-it | summ | cnn_dailymail | 1.0 |
| gemma-2-9b-it | vicuna-68m | coding | openai_humaneval | 1.0 |
| gemma-2-9b-it | vicuna-68m | long-ctx summ | scrolls | 0.99 |
| gemma-2-9b-it | vicuna-68m | summ | cnn_dailymail | 0.99 |
| phi-4 | Phi-3.5-mini-instruct | coding | openai_humaneval | 0.77 |
| phi-4 | Phi-3.5-mini-instruct | long-ctx summ | scrolls | 0.98 |
| phi-4 | Phi-3.5-mini-instruct | summ | cnn_dailymail | 0.99 |
| phi-4 | Qwen2.5-0.5B-Instruct | coding | openai_humaneval | 1.0 |
| phi-4 | Qwen2.5-0.5B-Instruct | long-ctx summ | scrolls | 1.0 |
| phi-4 | Qwen2.5-0.5B-Instruct | summ | cnn_dailymail | 1.0 |

Table 11: Results of injectivity tests for various tokenizers.

| Library | Tokenizer | Injective |
|---|---|---|
| SentencePiece | SentencePiece | True |
| Hugging Face | gpt2 | True |
| Hugging Face | double7/vicuna-68m | False |
| Hugging Face | bigcode/tiny_starcoder_py | True |
| Hugging Face | Qwen/Qwen2-0.5B-Instruct | True |

possible ways to concatenate adjacent tokens. Thus, the total number of valid concatenations is $2^{m-1}$, which follows from the combinatorial nature of partitioning the sequence into contiguous segments. $\qquad\square$

**Theorem 4.1.** *Let $p$ and $q$ be target and drafter probability distributions over vocabularies $T$ and $D$, respectively. Define $p', q_1, q_2$ to be probability distributions over $T \cup D$ as follows. $p'(x) = p(x)$ if $x \in T$ and $p'(x) = 0$ otherwise. $q_1(x) = q(x)$ if $x \in D$ and $q_1(x) = 0$ otherwise. $q_2(x) = \frac{q(x)}{\sum_{t \in T} q(t)}$ if $x \in T$ and $q_2(x) = 0$ otherwise. Given the target $p'$, we define $\alpha_1$ and $\alpha_2$ to be the probability of accepting a token $x \sim q_1$ and $x \sim q_2$, respectively, by the rejection sampling algorithm of speculative decoding from [Leviathan et al. (2023); Chen et al. (2023)]. Then, $\alpha_1 \leq \alpha_2$, and the output tokens distribute according to $p$.*

*Proof.* By Leviathan et al. (2023), the expected acceptance rate is the sum of the minimum probabilities of the target and draft distributions, namely, we have $\alpha_1 = \sum_{x \in T \cup D} \min\{p'(x), q_1(x)\} = \sum_{x \in T} \min\{p'(x), q_1(x)\} \leq \sum_{x \in T} \min\{p'(x), q_2(x)\} = \sum_{x \in T \cup D} \min\{p'(x), q_2(x)\} = \alpha_2$ since $\sum_{x \in T} q(x) \leq 1$. The output tokens distribute according to $p'$ because the rejection sampling algorithm of speculative decoding preserves the target distribution. Since $p'(x) = p(x)$ for $x \in T$, we have that the output tokens distribute according to $p$. $\qquad\square$

