# OpenReview forum: "Accelerating LLM Inference with Lossless Speculative Decoding Algorithms for Heterogeneous Vocabularies"
_ICML.cc/2025/Conference — ICML 2025 oral_

### Official Review · Reviewer_a7E7 · 2025-03-12

**Overall Recommendation:** 5

**Summary:**

This work proposes three methods for using a draft model with a different vocabulary than the target model in a typical speculative decoding framework. The authors propose: 1) string level exact matching (SLEM) in which the draft tokens are decoded back into string representations and reencoded by the target model tokenizer, 2) string level rejection sampling (SLRM) which modifies the above by including rejection sampling at the string level, and 3) token-level intersection (TLI) in which a modified draft vocabulary is sampled is generated as the intersection of target and original draft vocabulary, enabling more typical token-level verification from a subset of the original drafter vocabulary.  The authors explore numerous nuances and challenges that heterogeneous vocabularies introduce such as non-injective tokenizers, KV-cache considerations, lookahead controls which reduce unnecessary forward passes of the draft model. A small empirical study is conducted which compares autoregressive decoding to standard homogeneous SD and heterogeneous SD using SLEM and TLI.

**Claims And Evidence:**

Yes, in general the claims made are well supported by the discussion, illustrative examples, and proofs.

**Essential References Not Discussed:**

None noted.

**Experimental Designs Or Analyses:**

No issues noted. The experiments appear to be valid.

**Methods And Evaluation Criteria:**

Yes, the methods, datasets, and evaluation criteria used are generally standard. One area for improvement would be to include a standardized SD dataset such as SpecBench [1].
[1] https://github.com/hemingkx/Spec-Bench

**Other Comments Or Suggestions:**

* Algorithm 3 L6-8 could benefit from indentation or endif statements to clarify the conditional branch flow.
* Table 3: Consider adding “small vocabulary” to SLRS.
* L102: “draft token However”
* Suggest highlighting homogenous drafters in results as it’s not always clear which models share a vocab.

**Other Strengths And Weaknesses:**

## Strengths
* Important and timely topic.
* Original and novel approach to a practical challenge of implementing SD in practice.
* Well written
* Illustrative examples included in text
* Several target / draft pairs and datasets considered.
* Throughput increases are competitive with homogenous drafters.

## Weaknesses
* Additional figures to highlight the overall methods may be beneficial to the reader.
* Some select terms appear in text before definition, eg., lookahead value.
* No error bars / statistical analysis conducted on empirical results.
* “30 prompts” from the datasets used for Table 1 is somewhat informal and would be hard to reproduce. It would be best to use a standardized benchmark such as SpecBench.
* Additional discussion on the overhead for the SLEM method would be helpful. While we see throughput gains here I wonder about integration with more sophisticated inference engines such as vLLM. Could repeated tokenization/decoding block GPU in multi-tenant or high query per second settings?

**Questions For Authors:**

1. In my review of the supplementary materials I did not find the implementation for non-injective tokenizer “look behind”. Please confirm which section of the SM we can find this.
2. What are some challenges that may be encountered when implementing SLEM for vLLM or other inference engines which rely on asynchronous tokenization on CPU? Has any analysis been conducted on applying your proposed methods for multi-tenant or high query per second settings?

**Relation To Broader Scientific Literature:**

The challenge of producing bespoke draft models for each new potential target model is a big drawback when trying to use speculative decoding in practice. Some model families have natural draft candidates, such as Qwen2.5-0.5B for instance. However, even in this case the vocabulary for Qwen2.5-72B actually differs slightly from 0.5B (token IDs aligned but different size vocab.), highlighting the challenges of naively applying SD. Further, many organizations relying on SD may not have the necessary expertise, data, or compute to pretrain their own draft model or drafting heads. The proposed solutions and analysis in this work offer a unique, novel, and effective solution to these challenges. I am not aware of any other work that has tackled the heterogenous vocabulary problem in SD and as such consider this work seminal.

**Theoretical Claims:**

I briefly reviewed the proofs and they appear to be accurate.

---

> ### Author Rebuttal · Authors · 2025-03-27
>
> We are so grateful for your solid endorsement, rating our paper with the **highest score of 5 out of 5**! We are particularly thankful for your insightful acknowledgment of this work as a significant breakthrough:
> >“I am not aware of any other work that has tackled the heterogeneous vocabulary problem in SD … as such consider this work **seminal**.”
>
> We deeply appreciate your thoughtful recognition of our **“well written,” “important and timely,” “novel and effective solution” to a “big drawback when trying to use speculative decoding in practice.”** We also thank you for contributing a knowledgeable example of a real-world use case from your clearly strong hands-on experience, highlighting the importance of our solutions to the field-wide, genuine pain of practitioners applying speculative decoding:
> >“The challenge of producing bespoke draft models for each new potential target model is a big drawback when trying to use speculative decoding in practice. Some model families have natural draft candidates, such as Qwen2.5-0.5B for instance. However, even in this case the vocabulary for Qwen2.5-72B actually differs slightly from 0.5B (token IDs aligned but different size vocab.), highlighting the challenges of naively applying SD. Further, many organizations relying on SD may not have the necessary expertise, data, or compute to pretrain their own draft model or drafting heads.”
>
> We also thank you for reviewing our proofs and affirming their correctness. We truly appreciate your careful reading and for bringing to our attention some typos and suggested improvements in presentation, which have already helped improve the paper.
>
> ## A1. New Extended Benchmarks of SLEM and TLI
> Independently of our benchmarks, Hugging Face’s core maintainers have thoroughly evaluated the effectiveness of SLEM and TLI (Algorithms 2 and 4) and found our methods to be the most effective among all the speculative decoding algorithms they currently support. As a result, they made SLEM and TLI the default in Transformers (in Oct ’24 and Feb ’25, respectively), powering 5,000 other libraries with various use cases and hardware setups.
>
> To facilitate additional standardized benchmarks, we have open-sourced our benchmarking repository, which provides full reproducibility so anyone can compare our methods and any future alternatives on *exactly the same inputs and hardware*. We will attach a link to this repository upon publication.
>
> Furthermore, here are 2 extended benchmarks of SLEM and TLI, suggesting **up to 2.1× and 1.69× speedups** on various hardware setups: https://imgur.com/a/speculative-decoding-heterogeneous-vocabularies-extended-benchmark-of-algorithms-2-4-uV4PrTR (anon.)
>
> ## A2. Integrating SLEM and TLI into vLLM
> Thank you so much for your interest in integrating our algorithms into vLLM! Since SLEM and TLI have become the default of Hugging Face Transformers, we have received a lot of interest from users experiencing this pain who have asked about integrating them into vLLM.
>
> Thanks to vLLM’s support in disaggregated prefilling, the repeated process of SLEM is nonblocking and therefore should remain effective in both multi-tenancy and high query per second settings. Asynchronous tokenization is expected to increase the throughput in such setups.
>
> We do not see any theoretical constraints that would limit integration in vLLM or similar inference engines, nor do we see any major engineering gaps. In fact, we believe that vLLM will eventually support speculative decoding for heterogeneous vocabularies using these algorithms or future alternatives.
>
> ## A3. Implementation Details
> Thanks so much for your interest in SLEM’s implementation! The supplementary materials include the code we contributed to HF Transformers, which aligns with their naming conventions. Some naming differences exist, such as in the lookbehind logic of SLEM. The core logic is in the *AssistedCandidateGeneratorDifferentTokenizers* class, with the lookbehind mechanism implemented via a *diagonal matrix* referenced throughout the code. Helper functions like *_get_tokens_diag* appear only by signature and docstring. The full implementation is available on the *main* branch of HF Transformers.
>
> Regarding SpecBench, please note that while the implementations of SLEM and TLI allow homogeneous drafters, our primary focus is on heterogeneous drafters, in contrast to SpecBench, which benchmarks methods constrained to homogeneous drafters or self-speculation. Homogeneous methods are not applicable when the target lacks a model family (e.g., phi-4, Mixtral-8x22B). Also, homogeneous methods are ineffective if the smallest model in the family is still too slow, further limiting their applicability (e.g., see Figure 2-a in [1]). Some of the remaining methods in SpecBench are supported by HF Transformers and therefore were benchmarked in their independent experiments (see Section 1 above).
>
> ---
>
> Thank you for your support and feedback!
>
> [1]: arxiv.org/abs/2405.14105, ICLR ’25

---

### Official Review · Reviewer_7xEY · 2025-03-13

**Overall Recommendation:** 3

**Summary:**

The authors provide a comprehensive view of the challenges in performing speculative decoding with different vocabularies.
They come up with several solutions to address this, each with its own benefits and weaknesses.

In my personal experience, this is often a real headache, as training drafters for specific models is often difficult and time consuming.

**Claims And Evidence:**

-

**Essential References Not Discussed:**

-

**Experimental Designs Or Analyses:**

The authors also do a good job of showing specific examples and failure models of naive solutions, and coming up with several new methods.

However:

* For Algorithm 4, it would be nice to provide some example of the size of the intersection between some common tokenizers/vocabulary, to get a better sense of its usefulness
* The experiment section is somewhat lacking. The results are only computed for 30 prompt, which is really not much. I believe larger scales experiments are requirements, with more prompts/more seeds. Additionally, the speedups for most of the combinations in Table 1 either non-existent or not really impressive.
* There is also little explanation for why one drafter or one method would be better than another. It would be nice to see at least some heuristic to understand this better.

**Methods And Evaluation Criteria:**

The evaluation criteria (speedup for different draft/target configurations) make sense.

**Other Comments Or Suggestions:**

I am open to raising my score in light of some additional experimental results and explanations.

**Other Strengths And Weaknesses:**

The paper is very clear and well written.

**Questions For Authors:**

-

**Relation To Broader Scientific Literature:**

The contributions are, to my knowledge, novel and relevant in relation to the literature.

**Theoretical Claims:**

The exposition is rigorous, the algorithms are well motivated and the lossless-ness is proved.

---

> ### Author Rebuttal · Authors · 2025-03-28
>
> Thank you for your thoughtful review. We appreciate your recognition that our work provides **“a comprehensive view”** of speculative decoding with different vocabularies, that **“The exposition is rigorous, the algorithms are well motivated and the lossless-ness is proved,”** and that we do a **“good job of showing specific examples and failure models of naive solutions.”** We also value your observation that **“The contributions are, to my knowledge, novel and relevant in relation to the literature,”** and that **“The paper is very clear and well written.”** Moreover, we appreciate your personal insight—**“In my personal experience, this is often a real headache, as training drafters for specific models is often difficult and time consuming.”**—which underscores the pressing need to address heterogeneous vocabularies in speculative decoding.
>
> ### 1. Intersection Sizes are Provided
>
> You requested “some example of the size of the intersection between some common tokenizers/vocabulary”. Please note that:
> - Table 5 in Appendix C provides the sizes of the intersections for various target–drafter pairs.
> - Table 2 shows the expected acceptance rate for each pair, which governs its usefulness.
>
> We are eager to address all your concerns in hopes of justifying a higher score. Do you see any model pairs that we could add to Table 5 to significantly improve the paper? We are open to adding them all.
>
> ### 2. New Extended Benchmarks
>
> 1. Independently of our benchmarks, Hugging Face’s core maintainers have thoroughly evaluated the effectiveness of SLEM and TLI (Algorithms 2 and 4) and found our methods to be the most effective among all the speculative decoding algorithms they currently support. As a result, they made SLEM and TLI the default in Transformers (in Oct ’24 and Feb ’25, respectively), powering 5,000 other libraries with various use cases and hardware setups.
>
> 2. Section 1 of [our response to Reviewer a7E7](https://openreview.net/forum?id=vQubr1uBUw&noteId=6DDgLIbPNK) adds larger-scale benchmarks with additional pairs and hardware setups. These updated benchmarks suggest **significant speedups of up to 2.1× for SLEM and 1.69× for TLI**.
>
> 3. As [1] extensively studied, predicting the expected speedups of speculative decoding algorithms is possible by accurately estimating the ratio between the models’ forward latencies and acceptance rate. Table 2 provides the expected acceptance rate for all algorithms, and the ratio between the number of parameters of each model is often used as a surrogate for estimating the forward latencies ratio (see [2] for example).
>
> Our updated benchmarks above include 170 configurations of `<target, dataset, hardware, algorithm, drafter>`, each evaluated over 30 prompts, summing to a total of 5,100 runs. We designed these experiments to align with the highest standards of isolation, portability, and reproducibility, such that we completely sanitize the environment before each run—freeing all CPU and GPU memory. As a result, the initial setup of each run incurs a high overhead, especially because we must reload the models into the GPUs after clearing the memory, which can take a few minutes. This process requires reserving access to hardware for over a week when summing across all nodes and costs thousands of dollars, which is expensive given our constrained budget. Even if we had more budget to average over more prompts, since the set of all `<target, dataset, hardware, algorithm, drafter, prompt, seed>` is uncountable, any finite benchmark would still provide almost empty support.
>
> Nevertheless, we remain eager to address all your concerns to justify an even higher score. Given our new extended benchmarks, the acceptance rate analysis, the independent HF benchmarks, and the wide adoption in real-world software libraries for several months, what additional configurations could significantly improve the paper? We will do our best to accommodate specific requests within our limited budget toward a camera-ready version.
>
> ### 3. Choosing Drafters or Methods is Trivial
>
> In response to your request for “heuristics”, please note:
> - Table 2 reports the **exact expected acceptance rate** for each method, which—combined with the ratio of forward latency among the models—governs the overall efficiency, as analyzed in [1].
> - Our ‘Limitations’ section discusses at length the interplay between acceptance rates, forward latencies, and speedups.
>
> What does “heuristic” mean in this context? We always need to select drafters with the maximum acceptance rate and minimum forward latency.
>
> ---
>
> We truly value your input; your comments have already led to meaningful improvements in our experiments and exposition. **We sincerely appreciate your openness to raising your score in light of new experimental results, practical adoption by Hugging Face, and our explanations**.
>
> ---
>
> [1]: arxiv.org/abs/2405.14105, ICLR ’25
>
> [2]: arxiv.org/abs/2211.17192, ICML ’23

---

### Official Review · Reviewer_S2J2 · 2025-03-14

**Overall Recommendation:** 1

**Summary:**

This paper explores possible solutions for speculative decoding with a drafter model that does not share the vocabulary with the target model. Such methods, if successful, can enable the use of many more models as the drafter model for a large model to reduce the inference cost of large language models. The authors approached this problem with two distinct verification methods for speculative decoding: token match verification, and string match verification. Benchmarks demonstrated the success of the proposed methods when Gemma-2-9B-IT is used as the target model.

**Claims And Evidence:**

The main claims of this paper are the correctness and the effectiveness of the proposed algorithms.


The claim of correctness is reasonably substantiated,
-   The correctness of Algorithms 1, 3, and 4 can be easily established from the proof for the standard speculative decoding algorithm.
-   There is no explicit discussion on the correctness of Algorithm 2, but the correctness is also not difficult to prove.

However, additional data would be necessary to support the claim of effectiveness,
-   The benchmark results were reported in Table 1 in the main paper, and Table 7 in the appendix. Table 1 shows that with Gemma-2-9B-IT as the target model, Algorithms 2 & 4 can be faster than autoregressive decoding with vicuna-68m as the drafter, and sometimes even exceed the decoding speed of a Gemma-2-2B-IT drafter (same vocabulary as Gemma-2-9B-IT). However, Table 7 paints a different picture,
    -   There appears to be a trend that the proposed methods appear to not perform well on larger models: On quite a number of target/dataset combinations, the proposed algorithms underperforms or performs similarly to autoregressive decoding (e.g. Llama-3.1-8B-Instruct on CNN Daily Mail, Llama-3.1-70B on scrolls, Mixtral-8x22B-Instruct-v0.1 on scrolls and CNN Daily Mail, Llama-3.1-70B-Instruct on scrolls). While admitted the success of speculative decoding depends the choice of the drafter model, the prevalence of underperforming combinations calls into the question whether other factors are also contributing to the problem.
    -   Potential buggy implementation: The new token counts when temperature is 0 should be close, if not identical, across different methods on the same target model. However, they are drastically different in Llama-3.1-70B, CodeLlama-13b-Instruct-hf, and Llama-3.1-70B-Instruct.
-   Algorithm 3 was not benchmarked, and I am not confident that this algorithm can be made practical due to the need to iterate over a large number of possible drafter tokenizations.

**Essential References Not Discussed:**

None.

**Experimental Designs Or Analyses:**

The experimental design, which mostly involves benchmarks, is sound. However I have several doubts about the results and analysis of the effectiveness of the proposed methods. See my comments around Table 7 above.

**Methods And Evaluation Criteria:**

Yes.

**Other Comments Or Suggestions:**

-   Line 12 of Algorithm 5 should be "if j < i ... or else sample $t$ from $p_x()$".
-   Tables 4 & 5 are presented as part of the main paper but are in the appendix. I am confident that the authors will be able to find room for these tables after condensing their writing and removing unnecessary content.

**Other Strengths And Weaknesses:**

Strengths
-   The problem is well-motivated and successful solutions would be useful for LLM practitioners.
-   A large number of experiments were conducted to help readers evaluate the effectiveness of the proposed methods.
-   The proposed methods are simple modifications of existing methods, and thus are easy to understand and implement.

Weaknesses:

The presentation (especially organization and clarity) of this paper has a lot of room for improvement. For example,
-   Certain parts can be reordered to make the reading flow less disruptive to readers, e.g.the discussion of Algorithms 1 & 4 should be combined into a single Section 2.
-   "Algorithm 2 Supports Non-Injective Tokenizers" should be incorporated into the pseudocode of Algorithm 2. This is a crucial part for the correctness of this algorithm, however the current verbose description is both hard to follow and lacking crucial details (more on the second point later).
-   Many of the discussions in the text are confusing and unclear. For example,
    -   (099-104, left column): It appears to me that the condition $p(t) \leq q(t)$ for all $t \in T$ can only hold when $p = q$. Algorithm 1 being "optimal" in this case is not a terribly useful result. This paragraph seems to try to motivate Algorithm 4, but that can be done in a much simpler way by pointing out any samples from $D - T$ is useless.
    -   (082-089, right column): It appears to me that Algorithm 4 will sample from $T$ and thus not "sub-optimal". I don't understand why "we should accepted the token 'aa'".
    -   (111-114, right column): It appears to me that Algorithm 2 simply can't work if the vocabulary conditions do not hold. I don't understand why it would instead "leading to a decreased acceptance rate" if the acceptance rate is undefined in this context.
    -   (197-198, right column): "it does not guarantee that the output tokens are exactly the target tokens". I am confused about what "output tokens" means here as standard speculative decoding draws samples from the target distribution. What makes their samples not "the target tokens"? Or is it possible "output tokens" here really mean drafter sample tokens (which might get rejected)?
-   The writing can be made much less verbose. For example, (174-178, left column) could have been a simple sentence "$T \neq D$ limits the ability of the target and draft models to communicate tokeni IDs"; and the repeated reference to HuggingFace's wide adoption such as (040-042, right column) and the first paragraph of Section 5 could be just "our algorithms have already been implemented in the widely used HuggingFace transformers library".

**Questions For Authors:**

None.

**Relation To Broader Scientific Literature:**

This paper is a continuation on the works on using speculative decoding for improving LLM inference speed. It directly builds upon the seminal work on speculative decoding ([Leviathan et al, 2023], [Chen et al, 2023]), in both methods and proofs for correctness.

[Leviathan et al, 2023]: https://proceedings.mlr.press/v202/leviathan23a/leviathan23a.pdf
[Chen et al, 2023]: https://arxiv.org/pdf/2302.01318

**Theoretical Claims:**

I checked the proofs in Appendix E. They are correct as far as I can tell.

---

> ### Author Rebuttal · Authors · 2025-03-28
>
> Thank you for your thorough review and for underscoring several positive aspects of this work. We appreciate your noting that **“Benchmarks demonstrated the success of the proposed methods when Gemma-2-9B-IT is used as the target model,”** including the observation that **“Table 1 shows that with Gemma-2-9B-IT as the target model, Algorithms 2 & 4 can be faster than autoregressive decoding with vicuna-68m as the drafter, and sometimes even exceed the decoding speed of a Gemma-2-2B-IT drafter.”** We also value your statement that **“I checked the proofs in Appendix E. They are correct as far as I can tell.”** Furthermore, we appreciate your recognition that the proposed methods are **“easy to understand and implement,”** **“The experimental design, which mostly involves benchmarks, is sound”** and your emphasis on these:
> >**“Strengths**
> >- The problem is **well-motivated** and successful solutions would be **useful for LLM practitioners**.
> >- **A large number of experiments were conducted to help readers evaluate the effectiveness of the proposed methods.”**
>
> ---
>
> ### Improved Benchmarks Suggest Significant Speedups In Practice
>
> We've extended our benchmarks, as mentioned in [A1 for Reviewer a7E7
> ](https://openreview.net/forum?id=vQubr1uBUw&noteId=6DDgLIbPNK), and resolved the variance issue in the number of new tokens by filtering out crashed runs before averaging.
>
> We intentionally include configurations where our methods fail—to highlight their *limitations* instead of cherry-picking the best cases, communicating that practitioners should be careful when selecting heterogeneous drafters.
>
> SLEM and TLI are highly effective in practice and have been widely used in the industry during the past months, which indicates their real-world impact. The algorithms **do not “underperform”**. Like any SD algorithm, their effectiveness is controlled by:
> 1. Acceptance rates, as Table 2 provides.
> 2. Ratio between the models’ forward latencies.
>
> Larger targets often lead to acceleration, as you noticed. There's also an implementation overhead that has been shown to be negligible in light of the significant empirical speedups.
>
> ---
>
> >"There is no explicit discussion on the correctness of Algorithm 2, but the correctness is also not difficult to prove."
>
> The algorithm verifies that the final output string exactly matches the string that the target model generates, hence the correctness (losslessness) is derived immediately. What discussion do you believe is missing?
>
> >I don't understand why "we should accepted the token 'aa'".
>
> What advantage do you get from rejecting the draft token ‘aa’? The target model can only generate strings that contain the character ‘a’.
>
> >It appears to me that Algorithm 2 simply can't work if the vocabulary conditions do not hold. I don't understand why it would instead "leading to a decreased acceptance rate" if the acceptance rate is undefined in this context.
>
> The acceptance rate is always well-defined but might be zero.
>
> >"it does not guarantee that the output tokens are exactly the target tokens". I am confused about what "output tokens" means here as standard SD draws samples from the target distribution. What makes their samples not "the target tokens"? Or is it possible "output tokens" here really mean drafter sample tokens (which might get rejected)?
>
> SD algorithms operate on two distributions, given by probability vectors corresponding to the *drafter distribution* and *target distribution*. These algorithms output tokens, which effectively define an *output distribution*. Previous works proved that the output distribution aligns with the target distribution (also known as *losslessness*).
>
> The fact that two tokens are sampled from the same distribution **does not imply** they are equal, as stated in the paragraph that you mentioned.
>
> Thank you so much for asking this question; it has already helped us to improve the paper. We'll add an extended exposition on speculative decoding to enhance the clarity of future revisions.
>
> >Algorithm 3 was not benchmarked
>
> Please see A1: https://openreview.net/forum?id=vQubr1uBUw&noteId=gIxgR1GrKk.
>
> >Line 12 of Algorithm 5 should be "if $j < i$ ... or else sample $t$ from $p_x$".
>
> Thanks so much for bringing to our attention this issue. Beyond citing the papers that introduced SD, we included a rephrased version of their algorithm rather than copying it verbatim, as you noticed. Regarding your proposed change, please note that it samples the last token from $r_x$ rather than $p_x$ even if all the drafts are accepted (i.e., $j = i$), and hence **is lossy**.
>
> We corrected the mistake by editing line 12: Sample $t \sim r_x$ for $r_x(t):=\frac{p_x(t)-\min\{p_x(t),q_x(t)\}}{1-\sum_{t'\in T}\min\{p_x(t'),q_x(t')\}}$ if line 9 ever rejected a token. Otherwise, sample $t\sim p_x$.
>
> ---
>
> We are also truly grateful for your additional detailed suggestions for enhancing the ordering and presentation. They've already been very helpful in improving the paper.

---

### Official Review · Reviewer_8FKT · 2025-03-14

**Overall Recommendation:** 4

**Summary:**

This paper addresses a key limitation in existing speculative decoding (SD) methods for large language models (LLMs): the assumption that the drafter and target models share the same vocabulary. The authors propose three novel lossless SD algorithms—String-Level Exact Match (SLEM), String-Level Rejection Sampling (SLRS), and Token-Level Intersection (TLI)—which remove this constraint and support heterogeneous vocabularies. The proposed methods preserve the target distribution and work with off-the-shelf models, eliminating the need for costly retraining. The paper presents thorough theoretical guarantees and empirical evaluations across summarization, programming, and long-context tasks. Notably, one of the proposed methods (SLEM) has already been adopted as the default for heterogeneous SD in Hugging Face Transformers, demonstrating real-world impact.

**Claims And Evidence:**

Yes

**Essential References Not Discussed:**

N/A

**Experimental Designs Or Analyses:**

- **Lack of analysis of computational overhead in Algorithm 3:**
While Algorithm 3 (SLRS) is theoretically superior in acceptance rates, it may be impractical due to the exponential complexity of computing string probabilities ψ(t), especially for models with large vocabularies.

- **Limited Evaluation of Algorithm 3:**
The empirical evaluation focuses primarily on Algorithms 2 and 4. Although Algorithm 3 is interesting, its lack of experimental results makes it hard to judge its practical value.

**Methods And Evaluation Criteria:**

**Dependence on Vocabulary Overlap:**
Algorithm 4’s performance depends heavily on the intersection between the drafter and target vocabularies. In edge cases with low or no overlap, performance gains may vanish.

**Other Comments Or Suggestions:**

N/A

**Other Strengths And Weaknesses:**

**Strengths:**
1. **Novel Contribution:**
The work relaxes a long-standing constraint in speculative decoding—the requirement for vocabulary homogeneity—broadening its applicability significantly.

2. **Lossless and Theoretically Grounded:**
All proposed algorithms are rigorously proven to be lossless, with clear formal definitions, acceptance rate bounds, and theoretical guarantees (e.g., Theorems 3.1, 3.2, 4.1).

3. **Practical and Open Source Impact:**
The integration of Algorithm 2 into Hugging Face Transformers and its adoption as the default decoding method for heterogeneous vocabularies highlights immediate practical relevance and community validation.

**Weaknesses:**

See Above

**Questions For Authors:**

N/A

**Relation To Broader Scientific Literature:**

This idea has a broad impact in the era of speculative decoding

**Theoretical Claims:**

Yes

---

> ### Author Rebuttal · Authors · 2025-03-28
>
> Thank you for your detailed review and for highlighting so many strengths in our work. We appreciate your acknowledgment that **“this paper addresses a key limitation in existing speculative decoding,”** and that **“the paper presents thorough theoretical guarantees and empirical evaluations across summarization, programming, and long-context tasks.”** We also value your remark that **“this idea has a broad impact in the era of speculative decoding,”** and your observation that **“notably, one of the proposed methods (SLEM) has already been adopted as the default for heterogeneous SD in Hugging Face Transformers, demonstrating real-world impact.”** (– After this submission, TLI has also become the default in HF Transformers, as mentioned below.) Furthermore, we are glad you find **“Algorithm 3 is interesting”** and noted that **“Algorithm 3 (SLRS) is theoretically superior in acceptance rates.”** In particular, we are grateful that you recognize how these points underscore the significance of our contribution:
> >**“Strengths:**
> >- **Novel Contribution:** The work **relaxes a long-standing constraint** in speculative decoding—the requirement for vocabulary homogeneity—**broadening its applicability significantly**.
> >- **Lossless and Theoretically Grounded: All proposed algorithms are rigorously proven to be lossless, with clear formal definitions, acceptance rate bounds, and theoretical guarantees (e.g., Theorems 3.1, 3.2, 4.1).**
> >- **Practical and Open Source Impact: The integration of Algorithm 2 into Hugging Face Transformers and its adoption as the default decoding method for heterogeneous vocabularies highlights immediate practical relevance and community validation.”**
>
> Below, we address your concerns regarding the computational overhead of Algorithm 3 (SLRS) and the dependence of Algorithm 4 (TLI) on vocabulary overlap.
>
> ---
>
> ## A1. SLRS Proven Advantage
>
> The paper never claims that SLRS (algorithm 3) is practical *today*, with *existing* off-the-shelf models. Instead, we openly discuss its limitations in Section 3.4, where Lemma 3.3 transparently analyzes the tradeoffs of implementing SLRS with existing off-the-shelf models, which often have almost complete vocabularies (see ‘Vocabulary Constraints’ in Section 3.1 and Appendix D). Lemma 3.3 reveals how practitioners could design *new* vocabularies to facilitate SLRS.
>
> SLRS is mathematically proven to be:
> 1. Lossless (Theorem 3.2), as you mentioned in your review (“All proposed algorithms are rigorously proven to be lossless, with clear formal definitions, acceptance rate bounds, and theoretical guarantees”)
> 2. Increasing acceptance rates compared to Algorithms 1 and 2 (Table 2), as you mentioned in your review (“Algorithm 3 (SLRS) is theoretically superior in acceptance rates”)
>
> Your review mentions that SLRS is novel and addresses a long-standing key limitation. Therefore, we believe SLRS contributes to the research community by laying down theoretical foundations upon which future works can design vocabularies for heterogeneous drafters. Since heterogeneous drafters are a new research direction that has already shown potential (in this work and by its wide and quick adoption in practice) but has not been previously studied, such contributions could become fundamental.
>
> ## A2. TLI's Effectiveness Is Mathematically Proven + New Extensive Benchmarks Over Various Practical Setups
>
> The effectiveness of any speculative decoding algorithm is controlled by its acceptance rate, and TLI (Algorithm 4) is no exception.  In edge cases of low or no alignment between the draft and target distributions, all the known speculative decoding algorithms are expected to fail, including those in this paper.
>
> Table 2 provides the readers with the expected acceptance rate of all our algorithms, including TLI. We can see that the acceptance rate depends on the probability mass that the intersection supports. Please note that the acceptance rate is governed by the supported probability rather than by the size of the intersection (namely, the number of tokens in the intersection).
>
> Nevertheless, practitioners in the past years have often been using BPE, WordPiece, Unigram, or SentencePiece to construct tokenizers that share a reasonably large intersection, thanks to their heuristics (see ‘Vocabulary Constraints’ in Section 3.1 and Appendices C and D). In practice, TLI is highly effective in various setups and, therefore, has recently become the default behavior in Hugging Face Transformers after they conducted a thorough and independent benchmark. Please see [Section 1 of our response to Reviewer a7E7
> ](https://openreview.net/forum?id=vQubr1uBUw&noteId=6DDgLIbPNK) for details.
>
> ---
>
> Thanks so much for your support! We are eager to address any remaining concerns you might have.

---

> > ### Comment · Reviewer_8FKT · 2025-04-07
> >
> > Thanks for the clarification. I have raised my score. Good luck.

---

> > > ### Author Response · Authors · 2025-04-07
> > >
> > > **Thank you so much for raising your score to recommend *acceptance*!** We greatly appreciate your engagement in the discussion and are pleased that our previous response addressed your concerns.
> > >
> > > We remain eager to learn from your feedback and improve the paper further. What is the remaining concern that we have not fully addressed yet?
> > >
> > > Thank you again for your time and attention.

---

### Decision · Program_Chairs · 2025-05-01

**Decision:**

Accept (oral)

**Comment:**

This paper addresses a limitation of speculative decoding (SD) for large language models—namely, the assumption that the drafter and target model must share the same vocabulary. The authors introduce three lossless, training-free SD algorithms—SLEM, SLRS, and TLI—that decouple vocabulary requirements and maintain correctness of the target model’s output distribution. These methods offer varying trade-offs between theoretical optimality and practical speedup, and are applicable with off-the-shelf models and existing inference engines. The paper is well-motivated, technically solid, and shows clear potential for impact, both theoretically and practically.

A notable strength is the paper's real-world relevance and adoption: two of the proposed methods (SLEM and TLI) are already implemented as defaults in Hugging Face Transformers. The paper also provides provable guarantees and empirical evaluations across multiple LLMs.

Concerns:

- Reviewer 8FKT raised a concern that SLRS, while theoretically elegant, is currently too costly to deploy, and its utility remains largely conceptual.

- Reviewer 7xEY requests larger-scale benchmarks and more insight into when methods are effective.

These concerns are mostly satisfactorily addressed during rebuttal.

**Recommendation**. This paper is a strong example of research that combines theoretical insight with real-world relevance. It addresses a timely problem in LLM inference, removes a critical limitation in a widely used technique, and offers solutions that are both elegant and already influential in practice. I recommend acceptance.